# Generalized Dirichlet Energy and Graph Laplacians for Clustering Directed and Undirected Graphs

**Harry Sevi** [⋆†]                                           *harry.sevi@protonmail.com*

**Gwendal Debaussart-Joniec** [⋆†]             *gwendal.debaussart@ens-paris-saclay.fr*

**Malik Hacini** [⋆◇]                                       *malik.hacini@ens-paris-saclay.fr*

**Matthieu Jonckheere** [°]                           *matthieu.jonckheere@laas.fr*

**Argyris Kalogeratos** [⋆]                           *argyris.kalogeratos@ens-paris-saclay.fr*

[†] *Equal contribution.*

[⋆] *ENS Paris-Saclay, Université Paris-Saclay, CNRS, Centre Borelli, F-91190 Gif-sur-Yvette, France.*

[◇] *ENSIMAG, Grenoble INP, Université Grenoble Alpes, F-38400 Saint Martin d'Hères, France.*

[°] *CNRS, Laboratory for Analysis and Architecture of Systems, F-31400 Toulouse, France.*

**Reviewed on OpenReview:** *https://openreview.net/forum?id=AA6D7fJ9PN*

## Abstract

Clustering in directed graphs remains a fundamental challenge due to the asymmetry in edge connectivity, which limits the applicability of classical spectral methods originally designed for undirected graphs. A common workaround is to symmetrize the adjacency matrix, but this often leads to losing critical directional information. In this work, we introduce the *generalized Dirichlet energy* (GDE), a novel energy functional that extends the classical Dirichlet energy to handle arbitrary positive vertex measures and Markov transition matrices. GDE is a unified framework based on random walk diffusion dynamics, applicable to both directed and undirected graphs, and yields a family of generalized Laplacian matrices usable as drop-in operators in broader graph-learning pipelines. Building on GDE, we propose the *generalized spectral clustering* (GSC) method for clustering weakly connected digraphs without resorting to a random walk with teleportation. A key component of our approach is the utilization of a parametrized vertex measure encoding graph directionality and density. Experiments on real-world point-cloud and network datasets show that GSC consistently matches or outperforms existing spectral clustering methods in both clustering accuracy and robustness, making it a strong tool for graph-based data analysis.

**Keywords:** Dirichlet energy, random walks, graph Laplacian, parametrized graph operators, graph directionality, point-clouds, spectral clustering, graph partitioning.

## 1 Introduction

Clustering is a fundamental tool broadly used to uncover structure in large and complex datasets. In many modern applications, such as web graphs, citation networks, and information diffusion in social or transportation systems, a dataset is naturally represented as a directed graph (digraph), where edge direction encodes asymmetric relationships between data objects. Among the several graph clustering methods, spectral clustering has become popular due to its conceptual simplicity, efficiency, and strong theoretical underpinnings (Ng et al., 2002; Peng et al., 2015; Boedihardjo et al., 2021). A foundational concept in spectral clustering is the *Dirichlet energy*, which quantifies the smoothness of a function over its domain. For a graph function, defined over the vertices of a graph, this measures the variability of the function across adjacent vertices,

hence capturing local regularity with respect to graph connectivity. This notion underlies many successful clustering algorithms that have appeared in machine learning, network science, and other domains.

With data represented as a graph, graph clustering aims at partitioning that graph into subsets with high internal connectivity and sparse external links. This task is typically formulated as a discrete optimization problem, such as minimizing the graph cut or normalized cut, whose relaxed form corresponds to minimizing the Dirichlet energy of a graph function associated with a Laplacian operator (Von Luxburg, 2007; Shi & Malik, 2000). Spectral clustering solves this relaxed problem by computing the leading eigenvectors of the Laplacian, yielding low-dimensional embeddings that can be clustered via standard methods such as $k$-means.

Most spectral clustering methods are designed for undirected graphs. Many real-world systems, though, such as social and content networks or flow-based systems like transportation, are best modeled as digraphs carrying critical information in edge directionality. A widely adopted practice in handling weakly connected digraphs in graph machine learning tasks, such as spectral clustering (Zhou et al., 2005) or vertex classification (Peach et al., 2020), is the routine reliance on the teleporting random walk (Page et al., 1999). This process guarantees ergodicity, hence it is often used as a substitute for the natural random walk that may not be ergodic in directed graphs. Despite offering an alignment with classical ergodic-based theoretical frameworks, the teleporting random walk constitutes a convenient workaround rather than a principled modeling decision. Some of the drawbacks it introduces are: First, it adds uniform transitions to all vertices, as it blends the original graph structure with that of a complete graph. This alters its topology and injects non-local interactions, potentially masking critical structural information of the original graph. Imposing ergodicity in cases where it does not naturally hold can lead to misleading conclusions about the true graph dynamics (Schaub et al., 2019). Moreover, by densifying the graph it increases substantially the computational cost in practice. Aside from teleportation, other attempts have been made to address the lack of ergodicity. Zhou et al. (2005) build on the directed Laplacian of Chung (2005) to develop a spectral clustering method tailored to digraphs. Meilă & Pentney (2007) directly leverage the asymmetric adjacency matrix within a weighted cut framework. More recently, Rohe et al. (2016) proposed a spectral co-clustering approach based on a singular value decomposition, capturing directional information through low-rank approximations.

Point-cloud data commonly generate digraphs via $K$-nearest neighbor ($K$-NN) graphs, which are asymmetric by construction (Genton, 2002). Symmetrizing the adjacency matrix before clustering is standard (Satuluri & Parthasarathy, 2011), but this obscures directional patterns and can degrade performance.

A rather different line of work focuses on *flow-based clustering* (Cucuringu et al., 2020; Laenen & Sun, 2020; Coste & Stephan, 2021; Hayashi et al., 2022), aiming at identifying *imbalanced cuts*, i.e. partitions where most edges flow in one direction between clusters. This is particularly suited to domains with strongly asymmetric connectivity, such as migration networks (Cucuringu et al., 2020), food webs, or trade flows (Laenen & Sun, 2020). However, these approaches are distinct from the classical *density-based* clustering that seeks size-balanced clusters with high internal cohesion. More specifically, density-based clustering favors dense subgraphs and penalizes having low-degree or weakly connected vertices, while flow-imbalanced cuts look for sparse boundaries where only a small amount of 'flow' (edges) connects two subgraphs, regardless of how dense each side is internally.

In this work, we revisit the density-based clustering paradigm and we propose a novel principled framework that generalizes to both directed and undirected graphs. Our contributions are as follows:

- *Generalized Dirichlet energy* (**GDE**): We introduce a framework based on a novel energy functional that extends the classical Dirichlet energy (Montenegro et al., 2006). Unlike prior formulations, which are limited to ergodic random walks with specific transition matrices and stationary distributions, GDE is defined for *any* positive vertex measure and Markov transition matrix, without requiring reversibility or ergodicity. GDE stems naturally from a random walk interpretation of graph partitioning, and mitigates the issues appearing when the probability mass of the stationary distribution is concentrated unevenly across the graph. We also propose a parametrized vertex measure that encodes both directionality and edge density. The GDE framework and its associated family of generalized Laplacians are of general interest, as they can serve as drop-in graph operators in broader machine learning pipelines.

- *Generalized spectral clustering* (**GSC**): Building on the GDE, we develop a spectral clustering method that applies to both directed and undirected graphs. Unlike earlier spectral approaches that are limited to

strongly connected digraphs (Zhou et al., 2005; Palmer & Zheng, 2020), GSC applies to *weakly connected* digraphs without relying on teleporting random walks, such as those used in PageRank (Page et al., 1999). This leads to a principled and flexible clustering method suited for a wide variety of graph-structured data.

## 2 Preliminaries and background

### 2.1 Essential concepts

**Notation.** Let $\mathcal{G} = (\mathcal{V}, \mathcal{E}, w)$ be a weighted directed graph (digraph), where $\mathcal{V}$ is a finite vertex set of size $N = |\mathcal{V}|$, and $\mathcal{E} \subseteq \mathcal{V} \times \mathcal{V}$ is a set of directed edges. Each edge $(i, j)$ denotes a directed link from vertex $i$ to vertex $j$. Digraphs may be given directly or constructed from input point-clouds, i.e. from datapoints $\boldsymbol{X} = \{x_i\}_{i=1}^N$, with $x_i \in \mathbb{R}^d$, using appropriate graph construction techniques. We assume that $\mathcal{G}$ is a *weakly connected* digraph: that is, when the edge directionality is ignored, the obtained graph is connected.

The edge weight function $w : \mathcal{V} \times \mathcal{V} \to \mathbb{R}_+$ assigns a nonnegative value to each vertex pair: $w(i, j) > 0$ if $(i, j) \in \mathcal{E}$, and 0 otherwise. Let $\mathbf{W} = \{w_{ij}\}_{i,j=1}^N \in \mathbb{R}_+^{N \times N}$ be the adjacency matrix, where $w_{ij} = w(i, j)$. Define the out-degree and in-degree of vertex $i$ as $d_{\text{out}}(i) = \sum_j w_{ij}$ and $d_{\text{in}}(i) = \sum_j w_{ji}$, respectively. For a vector $\nu$, define $\mathbf{D}_\nu = \text{diag}(\nu)$. For a subset $S \subseteq \mathcal{V}$, its complement is $\bar{S} = \mathcal{V} \setminus S$, and its characteristic function is $\chi_S \in \{0, 1\}^N$, where $\chi_S(i) = 1$ iff $i \in S$. For a singleton set $S = \{v\}$, we use $\delta_v \in \{0, 1\}^N$, the Kronecker delta vector. Also, $\mathbf{1}_{N \times M}$ is the $N \times M$ all-ones matrix, and $\mathbb{1}\{\cdot\}$ is the indicator function.

A *positive vertex measure* $\nu : \mathcal{V} \to \mathbb{R}_+^*$ assigns strictly positive values to vertices; when $\sum_{i \in \mathcal{V}} \nu(i) = 1$, it is a *probability vertex measure*. A function $f : \mathcal{V} \to \mathbb{R}$ is represented as $f = [f(i)]_{i \in \mathcal{V}}^\mathsf{T} \in \mathbb{R}^N$. We assume graph functions belong to $\ell^2(\mathcal{V}, \nu)$, a Hilbert space with inner product:

$$\langle f, g \rangle_\nu = f^\mathsf{T} \mathbf{D}_\nu g = \sum_{i \in \mathcal{V}} \nu(i) f(i) g(i),$$

which reduces to the standard dot product $\langle f, g \rangle = f^\mathsf{T} g$ when $\nu = \mathbf{1}_{N \times 1}$. A *positive edge measure* is an edge-based function $q : \mathcal{V} \times \mathcal{V} \to \mathbb{R}_+$, constrained to $q(i, j) = 0 \iff (i, j) \notin \mathcal{E}$.

**Random walk fundamentals.** A natural random walk on $\mathcal{G}$ is defined by a homogeneous Markov chain $\mathcal{X} = (X_t)_{t \geq 0}$ on a state space $\mathcal{V}$, with transition probabilities:

$$p(i, j) = \mathbb{P}(X_{t+1} = j \mid X_t = i) = \frac{w(i, j)}{\sum_z w(i, z)}.$$

The transition matrix $\mathbf{P} = [p(i, j)] \in \mathbb{R}^{N \times N}$ is row-stochastic, and equals: $\mathbf{P} = \mathbf{D}_{\text{out}}^{-1} \mathbf{W}$. Its spectrum lies within the unit disk, i.e. all its eigenvalues have modulus at most 1. A stationary distribution $\pi$ is a probability vertex measure satisfying $\mathbf{P}^\mathsf{T} \pi = \pi$. Note that $\pi$ is not necessarily unique, and it may not be strictly positive (i.e. $\text{supp}(\pi) \subset \mathcal{V}$). In general, we simply refer to any such $\pi$ as *a* stationary distribution; when the stationary distribution is unique and strictly positive (i.e. $\text{supp}(\pi) = \mathcal{V}$), we say the random walk is *ergodic* (Brémaud, 2013; Seabrook & Wiskott, 2023). Ergodicity is verified if $\mathcal{G}$ is strongly connected and aperiodic, but it can also hold in other cases. When the random walk is ergodic, the distribution $p_t(i, \cdot) = \delta_i^\mathsf{T} \mathbf{P}^t$ converges to a unique stationary distribution $\pi \in \mathbb{R}_+^N$ as $t \to \infty$. For undirected graphs or reversible walks, the stationary distribution satisfies $\pi(i) \propto d(i)$, the degree of vertex $i$. Reversibility means $\pi(i) p(i, j) = \pi(j) p(j, i)$ for all $i, j$. This condition always holds in the undirected setting, where $d_{\text{out}}(i) = d_{\text{in}}(i) = d(i)$.

**Dirichlet energy and graph Laplacians.** The *Dirichlet energy* quantifies the global smoothness of a graph function, penalizing variation across edges. It is central in Dirichlet form theory (Saloff-Coste, 1997; Montenegro et al., 2006), graph signal processing (Shuman et al., 2013), and harmonic analysis on graphs (Sevi et al., 2023). Next, we give the random-walk-based definition of the Dirichlet energy.

**Definition 2.1 (Dirichlet energy of a graph function based on random-walks).** *Let $\mathcal{X}$ be a random walk on digraph $\mathcal{G}$ with transition matrix $\mathbf{P}$ and ergodic distribution $\pi$[1]. Then the Dirichlet energy of a*

---

[1]In the case of non-ergodic digraphs, the energy can be defined using a stationary distribution.

*function $f : \mathcal{V} \to \mathbb{R}$ is:*

$$\mathcal{D}^2(f) = \sum_{i,j \in \mathcal{V}} \pi(i) p(i,j) [f(i) - f(j)]^2 \tag{1}$$

$$= 2 \langle f, \mathbf{L}_{\mathrm{RW}} f \rangle_\pi = 2 \langle f, \mathbf{L} f \rangle. \tag{2}$$

Here, Eq. 1 encourages smooth functions over high-probability transitions. A feature of the random-walk-based formulation that we exploit later is that $\pi(i)$ reveals a global importance weight of vertex $i$. Eq. 2 links the Dirichlet energy to the Laplacian matrices $\mathbf{L}_{\mathrm{RW}}$, $\mathbf{L}$, $\overline{\mathbf{L}}$ (Chung, 2005; Sevi et al., 2023), which are fundamental to spectral methods.

**Definition 2.2** (**Directed Laplacians**). *Given a transition matrix $\mathbf{P}$ and its stationary distribution $\pi$, the directed Laplacians are defined as follows:*

$$\text{Random walk Laplacian:} \quad \mathbf{L}_{\mathrm{RW}} = \mathbf{I} - \tfrac{1}{2}(\mathbf{P} + \mathbf{D}_\pi^{-1} \mathbf{P}^\mathsf{T} \mathbf{D}_\pi) \tag{3}$$

$$\text{Unnormalized Laplacian:} \quad \mathbf{L} = \mathbf{D}_\pi - \tfrac{1}{2}(\mathbf{D}_\pi \mathbf{P} + \mathbf{P}^\mathsf{T} \mathbf{D}_\pi) \tag{4}$$

$$\text{Normalized Laplacian:} \quad \overline{\mathbf{L}} = \mathbf{D}_\pi^{-1/2} \mathbf{L} \mathbf{D}_\pi^{-1/2}. \tag{5}$$

In the particular case of a balanced digraph, where each vertex has equal in- and out-degree while generally $w(i,j) \neq w(j,i)$, we have that $\pi(i) \propto d_{\mathrm{out}}(i)$, thus the directed Laplacians reduce to the symmetric Laplacian of the symmetrized adjacency matrix (Seabrook & Wiskott, 2023):

$$\mathbf{L} = \mathbf{D}_{\mathrm{out}} - \tfrac{1}{2}(\mathbf{W} + \mathbf{W}^\mathsf{T}). \tag{6}$$

This is a natural extension of the undirected Laplacians to this setting. When the underlying graph is undirected, as it holds $\mathbf{W} = \mathbf{W}^\mathsf{T}$, we recover the classical undirected Laplacians $\mathbf{L} = \mathbf{D}_{\mathrm{out}} - \mathbf{W}$. In general, though, the directed Laplacians differ from their undirected counterparts and they cannot be obtained by simply symmetrizing the adjacency matrix.

## 2.2 Connection with spectral clustering

The Dirichlet energy yields a Laplacian operator, which is also a link to spectral methods. In spectral clustering, vertex embeddings are obtained from the eigenvectors of a chosen Laplacian. To formalize this connection, we recall the Courant-Fischer min-max theorem (Horn & Johnson, 1990) from spectral theory. For a Hermitian matrix $\mathbf{L}$ with eigenvalues $\lambda_1 \leq ... \leq \lambda_N$, the Rayleigh quotient:

$$\mathcal{R}_\mathbf{L}(y) = \frac{y^\mathsf{T} \mathbf{L} y}{y^\mathsf{T} y} = \frac{\langle y, \mathbf{L} y \rangle}{\|y\|^2}, \quad y \in \mathbb{R}^N \setminus \{0_N\}, \tag{7}$$

characterizes the eigenvalues via:

$$\lambda_k = \max_{\substack{M \subset \mathbb{R}^N \\ \dim(M) = N-k+1}} \min_{y \in M} \mathcal{R}_\mathbf{L}(y). \tag{8}$$

The $k$-th eigenvector, $v_k$, minimizes $\mathcal{R}_\mathbf{L}$, subject to orthogonality:

$$\lambda_k = \min_{y \in \mathbb{R}^N} \mathcal{R}_\mathbf{L}(y), \quad v_k = \arg\min_{y \in \mathbb{R}^N} \mathcal{R}_\mathbf{L}(y) \tag{9}$$

$$\text{subject to } \langle v_j, y \rangle = 0, \text{ for } j = 1, ..., k-1. \tag{10}$$

**Remark 2.1.** *The Dirichlet energy and the Rayleigh quotient are proportional:*

$$\overline{\mathcal{D}}^2(f) = 2 \mathcal{R}_\mathbf{L}(f), \quad \Rightarrow \quad \min_{f \in \mathbb{R}^N} \mathcal{R}_\mathbf{L}(f) \equiv \min_{f \in \mathbb{R}^N} \overline{\mathcal{D}}^2(f), \tag{11}$$

*where $\overline{\mathcal{D}}^2(f) = \frac{\mathcal{D}^2(f)}{\|f\|^2}$ is the* normalized Dirichlet energy.

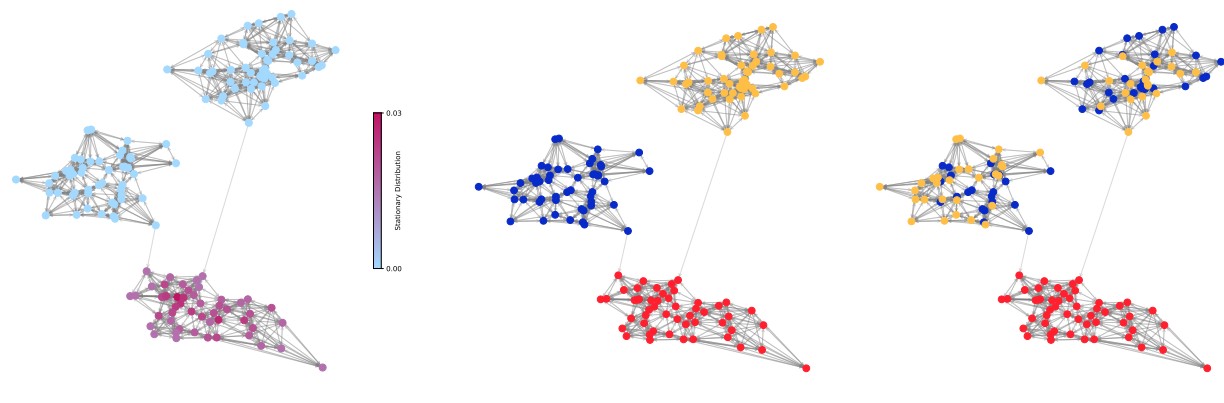

(a) Stationary distribution $\pi$         (b) Two partitions minimizing the classical Dirichlet energy

Figure 1: **Weak connectivity and the Dirichlet energy.** a) The heterogeneous stationary distribution $\pi$ in an unweighted digraph with three clusters, where the bottom one is a 'sink' (i.e. it has only incoming edges). In this case, the support of $\pi$ does not cover all the vertex set. b) Two partitions $f$, $g$, both minimizers of the classical Dirichlet energy, i.e. $\mathcal{D}^2(f) = \mathcal{D}^2(g)$. The left one captures the cluster structure, while the other mixes the clusters that receive zero mass from the stationary distribution, which is the reason why the Dirichlet energy is insensitive to function variations in those areas.

Therefore, minimizing the Rayleigh quotient is equivalent to minimizing the Dirichlet energy. As minimizing the Dirichlet energy promotes smooth functions over a graph, spectral clustering assigns similar values to well-connected vertices. In particular, for two clusters ($k = 2$), one may use a binary function of the form $f = \chi_{V_1}$, and here more specifically the characteristic function of the set $V_1$. After relaxation, i.e. allowing $f$ to be a non-binary function, minimizing the Dirichlet energy is equivalent to finding the second eigenvector (Fiedler vector) of $\mathbf{L}$. The second eigenvector can then be used to find a 2-partition of the graph, as it minimizes the Dirichlet energy while being orthogonal to the constant vector. This generalizes naturally to multi-way partitioning (see Section 4).

## 3 The generalized Dirichlet energy framework

While the classical Dirichlet energy is a powerful tool for graph-based learning, it has limitations, notably when applied to digraphs, especially weakly connected ones. In this section, we first discuss these limitations and then introduce the *generalized Dirichlet energy* (GDE), which extends the classical formulation to accommodate arbitrary positive vertex measures and Markov transition matrices. GDE allows capturing a wider range of graph structures and dynamics, particularly in settings where the stationary distribution of a random walk may not be well-defined or may exhibit undesirable properties. We show that GDE gives rise to a family of self-adjoint generalized graph Laplacians, which can be used to develop spectral clustering methods that are applicable to both directed and undirected graphs, without relying on the ergodicity of the underlying random walk.

### 3.1 Motivation and intuition

In the classical formulation of graph segmentation based on random walks, the Dirichlet energy is defined with respect to the stationary distribution $\pi$ of the random walk. This approach works well for undirected graphs or strongly connected digraphs where $\pi$ is well-defined and captures the long-term behavior of the walk. However, in weakly connected digraphs, the random walk is not ergodic, and $\pi$ cannot be properly defined. In this context, ergodic means that there is a *unique* stationary distribution with a full support over the vertex set, i.e. $\text{supp}(\pi) = \mathcal{V}$ and thus having positive weight for all vertices. Note however that one (or more) stationary distribution with partial support over a vertex subset, i.e. $\text{supp}(\pi) = \mathcal{V}' \subset \mathcal{V}$, might

still exist (Seabrook & Wiskott, 2023). Such stationary distributions break the classical Dirichlet energy framework, as they would zero out the energy contribution of vertices outside the support $\mathcal{V}\backslash\mathcal{V}'$.

Fig. 1 shows an example of a weakly connected digraph featuring a stationary distribution with partial support. Any graph functions $f$ and $g$ such that $f(i) = g(i)$ for all $i \in \mathcal{V}'$ and $f(i) \neq g(i)$ for some $i \in \mathcal{V}\backslash\mathcal{V}'$ would lead to $\mathcal{D}^2(f) = \mathcal{D}^2(g)$, since the energy contribution of the vertices in $\mathcal{V}\backslash\mathcal{V}'$ is zero. In the context of clustering, this suggests that partitions assigning different cluster labels to vertices in $\mathcal{V}\backslash\mathcal{V}'$ would be indistinguishable when looking at the Dirichlet energy, regardless of how well they match with the underlying cluster structure of the graph. For instance, in Fig. 1b, both partitions minimize the Dirichlet energy, but only the left one captures the natural cluster structure of the graph, while the right one mixes clusters that receive zero mass from the stationary distribution.

Similar concerns arise even for undirected graphs in the presence of degree heterogeneity, since high-degree vertices can dominate the energy landscape by overshadowing low-degree vertices, and thus lead to biased clustering results. Indeed, the Dirichlet energy is a sum of vertex-wise contributions weighted by $\pi$:

$$\mathcal{D}^2(f) = \sum_{i,j \in \mathcal{V}} \pi(i)p(i,j)[f(i)-f(j)]^2 = \sum_{i \in \mathcal{V}} \pi(i) \sum_{j \in \mathcal{V}} p(i,j)[f(i)-f(j)]^2. \tag{12}$$

This tells us that, even if $\text{supp}(\pi) = \mathcal{V}$, when minimizing the Dirichlet energy, a higher $\pi(i)$ induces a higher penalization of function variation at vertex $i$. Therefore, the distribution of the mass of $\pi$ across the graph may heavily affect which parts of the cluster structure are easier to detect. For instance, the high concentration of the mass at a few individual vertices, or specific clusters, would make the Dirichlet energy more sensitive to function variations at those regions, while hindering the correct detection of the cluster structure in the rest of the graph. Intuitively, $\pi$ carries global information about the significance of each vertex for the graph; however, when $\pi$ exhibits high heterogeneity it is reasonable to seek instead a 'smoother' surrogate of that measure with a flatter shape; this aspect is developed further in Section 3.3.

## 3.2   Generalized Dirichlet energy and graph Laplacians

We now formalize the connection between Dirichlet energy and graph Laplacians through the introduction of the *generalized Dirichlet energy* (GDE). This functional is defined with respect to a general positive edge measure $q$, and naturally leads to a broader class of graph Laplacians. We begin with the general definition of GDE, then specialize to cases where $q$ is derived from a vertex measure $\nu$ and a transition matrix $\mathbf{P}$. The derived family of Laplacian operators adapts to the graph structure as seen through the random walk dynamics. Although our focus is spectral clustering, the operators defined through the GDE framework can be of use in other graph-based learning problems.

**Definition 3.1** (**Generalized Dirichlet Energy of a graph function**). *Let $q : \mathcal{V} \times \mathcal{V} \to \mathbb{R}_+$ be a positive edge measure on a digraph $\mathcal{G}$, and let $\mathbf{Q} = [q(i,j)]_{i,j \in \mathcal{V}}$ denote the corresponding matrix. The generalized Dirichlet energy of a function $f : \mathcal{V} \to \mathbb{R}$ is given by:*

$$\mathcal{D}^2_{\mathbf{Q}}(f) = \sum_{i,j \in \mathcal{V}} q(i,j)[f(i)-f(j)]^2. \tag{13}$$

This formulation is generic as the graph structure is captured by the choice of edge measure $q$. Most typically, the edge weights of the input graph are directly plugged in by setting $q(i,j) = w(i,j)$, which –from a minimization point-of-view– essentially encourages smooth functions over edges with high weights. Another instance of the energy functional of Eq. 13 is through the lens of a random walk, by setting $q(i,j) = \pi(i)p(i,j)$:

$$\mathcal{D}^2_{\mathbf{D}_\pi \mathbf{P}}(f) = \sum_{i,j \in \mathcal{V}} \pi(i)p(i,j)[f(i)-f(j)]^2 = \mathcal{D}^2(f), \tag{14}$$

where $\pi$ is the stationary distribution of a random walk with transition matrix $\mathbf{P}$. This yields the classical random-walk-based Dirichlet energy form ($\mathcal{D}^2(f)$) that we saw earlier in Definition 2.1.

An important feature of the random-walk-based formulation is that it decomposes the edge measure as $q(i,j) = \pi(i)p(i,j)$: $\pi(i)$ expresses the explicit global importance of vertex $i$ as the probability for the random

walk to be at that vertex anytime (see Eq. 12 and the discussion in Section 3.1), and $p(i,j)$ is a local weight given that the walk is already at vertex $i$. Relying on this aspect, we use the form $q(i,j) = \nu(i)p(i,j)$, with $\nu$ being a tunable positive vertex measure, and obtain the GDE form:

$$\mathcal{D}_\nu^2(f) = \sum_{i,j \in \mathcal{V}} \nu(i)p(i,j)[f(i) - f(j)]^2. \tag{15}$$

This enables the definition of new energy functionals with arbitrary positive vertex measures $\nu$. Since scaling $\nu$ simply rescales the energy by $\|\nu\|_1^{-1}$, we may also assume without loss of generality that $\nu$ is a probability vertex measure.

**Definition 3.2** (**Generalized graph Laplacians**). *Let $\mathcal{X}$ be a random walk on a digraph $\mathcal{G}$ with transition matrix $\mathbf{P}$. Let $\nu$ be a positive vertex measure, and define the incoming measure $\xi = \mathbf{P}^\mathsf{T}\nu$. Define $\mathbf{D}_{\nu+\xi} = \mathbf{D}_\nu + \mathbf{D}_\xi$. The associated generalized Laplacians are:*

$$\text{Unnormalized:} \quad \mathbf{L}_\nu = \mathbf{D}_{\nu+\xi} - (\mathbf{D}_\nu\mathbf{P} + \mathbf{P}^\mathsf{T}\mathbf{D}_\nu), \tag{16}$$

$$\text{Random walk:} \quad \mathbf{L}_{\mathrm{RW},\nu} = \mathbf{D}_{\nu+\xi}^{-1}\mathbf{L}_\nu, \tag{17}$$

$$\text{Normalized:} \quad \overline{\mathbf{L}}_\nu = \mathbf{D}_{\nu+\xi}^{-1/2}\mathbf{L}_\nu\mathbf{D}_{\nu+\xi}^{-1/2}. \tag{18}$$

Here, $\xi = \mathbf{P}^\mathsf{T}\nu$ corresponds to the Perron–Frobenius operator applied to $\nu$ (Ding & Zhou, 2009; Klus & Trower, 2023; Klus & Djurdjevac Conrad, 2023). When $\nu = \pi$, we recover classical Laplacians:

$$\mathbf{L}_\pi = 2\mathbf{D}_\pi - (\mathbf{D}_\pi\mathbf{P} + \mathbf{P}^\mathsf{T}\mathbf{D}_\pi) = 2\mathbf{L}. \tag{19}$$

**Proposition 3.1.** *Let $\nu$ be a positive vertex measure and $\xi = \mathbf{P}^\mathsf{T}\nu$. Then for any graph function $f$:*

$$\mathcal{D}_\nu^2(f) = \langle f, \mathbf{L}_{\mathrm{RW},\nu}f\rangle_{\nu+\xi} = \langle f, \mathbf{L}_\nu f\rangle.$$

*Moreover, the Dirichlet energy can be expressed as follows:*

$$\mathcal{D}_\nu^2(f) = \frac{1}{2}\sum_{i,j \in \mathcal{V}} \left(\nu(i)p(i,j) + \nu(j)p(j,i)\right)[f(i) - f(j)]^2.$$

This implies that the generalized Dirichlet energy can be expressed as a quadratic form of the generalized Laplacian $\mathbf{L}_\nu$ in the standard inner product. It also implies that the Laplacians are positive semi-definite and thus admit a real eigenvalue decomposition. This is formalized in the following property.

**Proposition 3.2.** *The unnormalized and normalized generalized Laplacians, $\mathbf{L}_\nu$ and $\overline{\mathbf{L}}_\nu$, are self-adjoint and positive semi-definite on $\ell^2(\mathcal{V})$. The generalized random walk Laplacian, $\mathbf{L}_{\mathrm{RW},\nu}$, is self-adjoint on $\ell^2(\mathcal{V}, \nu+\xi)$.*

***Rayleigh quotient.*** The normalized generalized Dirichlet energy is given by:

$$\overline{\mathcal{D}}_\nu^2(f) = \frac{\mathcal{D}_\nu^2(f)}{\|f\|^2}. \tag{20}$$

This aligns with the Rayleigh quotient principle, as we have:

$$\mathcal{R}_{\mathbf{L}_\nu}(f) = \frac{\langle f, \mathbf{L}_\nu f\rangle}{\langle f, f\rangle} = \overline{\mathcal{D}}_\nu^2(f).$$

This connection allows us to leverage spectral theory to analyze the properties of the generalized Laplacians, and to develop spectral clustering methods based on the GDE framework.

## 3.3 Parametrized vertex measure

While the stationary distribution $\pi$ captures the long-term behavior of a random walk, which summarizes important information about the graph structure, it may not be the best choice for the vertex measure in the

Dirichlet energy when $\pi$ is highly heterogeneous (Section 3.1). To address this, we propose vertex measures derived from random walk dynamics that interpolate between the uniform distribution (treating all vertices equally) and the stationary distribution (capturing long-term behavior), providing a more balanced graph representation for clustering.

***Vertex measure constructions.*** In this part, we construct vertex measures, denoted by $\nu$, that capture local and global graph dynamics via random walks. Starting from a uniform measure, we define the following family of measures that depend on the diffusion time $t$, which generally needs to be short, as well as on the choice of the initializer vertex measure $\mu$:

$$\nu_t = [\mathbf{P}^t]^\mathsf{T}\mu. \tag{21}$$

**Proposition 3.3.** *Let $\nu_t = [\mathbf{P}^t]^\mathsf{T}\mu$, where $\mathbf{P}$ is ergodic and $\mu$ is a probability vertex measure acting as the distribution that initializes the random walk. Then, $\nu_t \xrightarrow{t\to\infty} \pi$ and:*

$$\lim_{t\to\infty} \mathcal{D}^2_{\nu_t}(f) = \mathcal{D}^2_\pi(f) = \mathcal{D}^2(f). \tag{22}$$

Note that for any time-dependent vertex measure $\nu_t$ such that $\nu_t \xrightarrow{t\to\infty} \nu$, it holds $\mathcal{D}^2_{\nu_t}(f) \xrightarrow{t\to\infty} \mathcal{D}^2_\nu(f)$. The original Dirichlet energy (Eq. 1) operates in the stationary regime and uses $\mu = \pi$ as an initializing measure; by the definition of $\pi$, this would yield $\nu_t = [\mathbf{P}^t]^\mathsf{T}\pi = \pi$. However, as explained earlier, using $\pi$ as vertex measure in the energy functional can be problematic when $\pi$ exhibits high heterogeneity across the graph, or when it is concentrated on a subset of vertices. To alleviate this problem, we propose the use of the uniform measure $\mu = \frac{1}{N}\mathbf{1}_{N\times 1}$ as initializer to treat all vertices fairly, along with a short random walk (small $t$) that extracts important structural information about the graph without allowing the $\nu_t$ measure to get too close to $\pi$. In simple terms, this approach aims to construct a flattened surrogate of the $\pi$ distribution. The time parameter $t$ controls measure locality and relates closely to mixing times (Levin & Peres, 2017).

On top of the base measure $\nu_t$ initialized by $\mu = \mathbf{1}/N$, we can define the following more flexible parametrized measure: given $t \in \mathbb{N}$ and $\alpha \in \mathbb{R}_+$,

$$\nu_{t,\alpha} = \left([\mathbf{P}^t]^\mathsf{T} \tfrac{1}{N}\mathbf{1}_{N\times 1}\right)^{\odot\alpha}, \quad \text{i.e.}\, \nu_{t,\alpha}(i) = \left([\mathbf{P}^t]^\mathsf{T} \tfrac{1}{N}\mathbf{1}_{N\times 1}\right)^\alpha_i. \tag{23}$$

The proposed family of vertex measures $\nu_{t,\alpha}$ is grounded in two key principles. First, the distribution $\nu_{t,\cdot}$ reflects the outcome of a $t$-step diffusion starting from a uniform distribution[2]. For reversible and ergodic random walks, as $t \to \infty$, the underlying diffusion converges to $\pi$, and therefore $\nu_{t,1} \to \pi$. Second, the exponent $\alpha$ acts as a sharpness parameter, inspired by transformations used in normalized Laplacians and information-theoretic re-weightings. Varying $\alpha$ allows controlling the distribution's concentration: $\alpha < 1$ flattens the measure, $\alpha > 1$ sharpens it, and $\alpha = 0$ makes the measure uniform (with the convention that $0^0 = 1$). This formulation offers greater flexibility to adapt to graph heterogeneity. Note that setting $\alpha \neq 1$ gives a non-probability measure, which however does not affect the essence or validity of the model.

Plugging the proposed $\nu_{t,\alpha}$ into Eq. 15 yields:

$$\mathcal{D}^2_{\nu_{t,\alpha}}(f) = \langle f, \mathbf{L}_{t,\alpha}f\rangle, \tag{24}$$

where $\mathbf{L}_{t,\alpha}$ is the generalized Laplacian associated with the reweighted measure $\nu_{t,\alpha}$. The proposed $\nu_{t,\alpha}$ enables smooth transitions between local/global and sharp/flat configurations via the computation of a family of operators $(\mathbf{L}_{t,\alpha})_{t,\alpha}$. This family of operators supports adaptive, data-driven spectral clustering, with an emphasis on flexibility.

Tuning $t$ and $\alpha$ in an unsupervised manner can be challenging, and if not done carefully may affect the quality of the resulting clustering. When reliable validation criteria are unavailable, a practical alternative is to resort to parameter-free vertex measures that remove this dependency altogether. Natural choices include the

---

[2]Zero-support entries make the definition of generalized Laplacians intractable. This occurs when certain vertices are not reachable by the random walk from any other vertex (e.g. in the presence of source vertices). In practice, those entries are set to a small positive value.

degree measure $\nu \propto d_{\text{in}} + d_{\text{out}}$[3], the uniform measure $\nu \propto \mathbf{1}$, or the Perron vector of $\mathbf{P}$ i.e. $\nu = \mathbf{P}^{\mathsf{T}}\nu$, although, as specified in Section 3.1 this measure can be supported only on a subset of vertices. Such alternatives, along with their properties and practical trade-offs, are discussed in Section B.

## 4 Generalized spectral clustering via GDE minimization

The *generalized spectral clustering* (GSC) method, based on GDE minimization, is applicable to any graph, directed or not. The method stems from the random walk viewpoint of graph partitioning. We start by revisiting the random walk perspective of graph 2-partitioning and then move to $k$-partitioning.

### 4.1 GDE of a graph 2-partition

For an arbitrary probability vertex measure $\nu : \mathcal{V} \to \mathbb{R}^*_+$, let $\nu(S)$ be its evaluation over a subset $S \subseteq \mathcal{V}$: $\nu(S) = \sum_{i \in S} \nu(i)$. Let also $q : \mathcal{E} \to \mathbb{R}_+$ be the composite edge measure such that $q(i,j) = \nu(i)p(i,j)$. Respectively, we can define $q(S,U)$ to be the edge measure between two disjoint vertex subsets $S, U \subseteq \mathcal{V}$, $S \cap U = \emptyset$, and generally $S \cup U \neq \mathcal{V}$:

$$
\begin{aligned}
q(S,U) &= \sum_{i \in S, j \in U} q(i,j) = \sum_{i \in S, j \in U} \nu(i)p(i,j), \\
&= \sum_{i \in S, j \in U} \mathbb{P}(X_t = i)\mathbb{P}(X_{t+1} = j \mid X_t = i), \\
&= \mathbb{P}(X_t \in S, X_{t+1} \in U), \quad \text{for any } t \geq 0.
\end{aligned}
\tag{25}
$$

The generic measure $q(S,U)$ is related to an ergodic Markov chain at the equilibrium, i.e. when $\nu = \pi$ (Sinclair, 1992; Levin & Peres, 2017). We generalize $q(S,U)$ for any transition matrix and vertex measure to quantify the escape probability from $S$ to $U$ in one step, when the starting vertex is drawn from the measure $\nu$. The case $U = \bar{S}$ is particularly relevant for graph partitioning. In essence, $q(S,\bar{S})$ offers a probabilistic point of view over the graph cut between the set $S$ and the rest of the graph.

Note that Meilă & Shi (2001) have presented a similar measure in the ergodic setting $\nu = \pi$. Next, we establish the connection between GDE and the edge measure $q(S,\bar{S})$ (the proof is in Section C.4).

**Proposition 4.1.** *Let $\mathcal{X}$ be a random walk on a digraph $\mathcal{G}$, with transition matrix $\mathbf{P}$. Let $\nu$ be a positive vertex measure, and $q$ be its associated positive edge measure, both on $\mathcal{G}$. Let $S \subseteq \mathcal{V}$ and $\bar{S} = \mathcal{V} \backslash S$. Consider the characteristic function $\chi_S$, associated with the set $S$, as a graph function. The composite edge measure $q(S,\bar{S})$ and the GDE are related as follows:*

$$
q(S,\bar{S}) + q(\bar{S},S) = \mathcal{D}^2_\nu(\chi_S).
\tag{26}
$$

In the case of a 2-partitioning, the function $\chi_S$ serves as an indicator (partition) function that separates the vertex set into two disjoint parts, $S$ and $\bar{S}$ (see also Section 2.2). Eq. 26 thus provides a natural interpretation of the generalized Dirichlet energy for graph partitions: the quantity $\mathcal{D}^2_\nu(\chi_S)$ measures how easily a random walk with transition probabilities $q$ escapes from $S$ to $\bar{S}$, or vice versa. This energy is symmetric by construction, since $\mathcal{D}^2_\nu(\chi_S) = \mathcal{D}^2_\nu(\chi_{\bar{S}})$. Under mild assumptions[4], one can further show that $\mathcal{D}^2_\nu(\chi_S)$ corresponds to the normalized cut (N-Cut) of a graph with edge weights $q(i,j)$. In this context, the vertex measure $\nu$ plays the role of a regularizer in the graph cut objective, shaping the structure of the optimal partition.

When the vertex measure is chosen as the stationary distribution $\nu = \pi$ of the random walk, the following corollary holds (the proof is in Section C.5).

**Corollary 4.1** (of Proposition 4.1). *Let $\mathcal{X}$ be an ergodic random walk on a digraph $\mathcal{G}$, with transition matrix $\mathbf{P}$ and ergodic (stationary) distribution $\pi$. For any subset $S \subseteq \mathcal{V}$, let $\bar{S} = \mathcal{V} \setminus S$, and let $\chi_S$ be the indicator*

---

[3]Due to the constraint $\nu(i) > 0$, the degree measure should include both in- and out-degrees to be well-defined in weakly-connected graphs.

[4]The graph defined by the transition matrix $q(i,j)$ must be connected and aperiodic (non-bipartite).

*function of $S$. Then, the composite edge measure $q(S, \bar{S})$ associated with $\pi$ and the generalized Dirichlet energy are related as:*

$$q(S, \bar{S}) = \frac{1}{2}\mathcal{D}_\pi^2(\chi_S).$$

This corollary reveals a key insight: minimizing the Dirichlet energy $\mathcal{D}_\pi^2(\chi_S)$ is equivalent to minimizing the total ergodic flow crossing between $S$ and its complement $\bar{S}$. In other words, $\mathcal{D}_\pi^2(\chi_S)$ serves as a normalized cut-like objective under the stationary distribution of the random walk. More generally, in this work we establish the idea that if a probability measure $\nu$ is used in the place of $\pi$ (e.g. a measure that approximates or regularizes $\pi$) then minimizing $\mathcal{D}_\nu^2(\chi_S)$ can still be interpreted as finding a well-separated 2-partition of the graph under the dynamics induced by $\nu$.

### 4.2 GDE minimization for graph $k$-partitioning

The goal of graph $k$-partitioning is to divide the vertex set of a digraph into $k$ disjoint subsets such that the edge density across subsets is minimized. Let $\boldsymbol{V} = \{V_\kappa\}_{\kappa=1}^k$ denote such a partition, where each $V_\kappa \subset \mathcal{V}$ is a subset of the vertex set of the graph, and $V_\kappa \cap V_{\kappa'} = \emptyset$ for $\kappa \neq \kappa'$, with $\bigcup_{\kappa=1}^k V_\kappa = \mathcal{V}$. Associated with each subset $V_\kappa$, there is a partition function $\chi_{V_\kappa} \in \{0,1\}^N$ that serves as an indicator vector over the vertices. The partition functions define a set of binary signals $\{\chi_{V_\kappa}\}_{\kappa=1}^k$, each highlighting one cluster versus the rest.

We define the *Dirichlet energy of the $k$-partition* under a vertex measure $\nu$ as:

$$\mathcal{D}_\nu^2(\boldsymbol{V}) = \sum_{\kappa=1}^k \mathcal{D}_\nu^2(\chi_{V_\kappa}), \tag{27}$$

where each term measures the generalized Dirichlet energy associated with a single cluster.

Let $\mathbf{U} = [u_1\ u_2\ \cdots\ u_k] \in \mathbb{R}^{N \times k}$ be the matrix whose columns are the indicator vectors $u_\kappa = \chi_{V_\kappa}$. The generalized spectral clustering (GSC) method, based on the GDE framework, casts the graph partitioning task as the following optimization problem:

$$\min_{\boldsymbol{V} = \{V_1, \ldots, V_k\}} \mathcal{D}_\nu^2(\boldsymbol{V}) = \min_{\mathbf{U}} \ \mathrm{tr}(\mathbf{U}^\mathsf{T}\mathbf{L}_\nu\mathbf{U}) \quad \text{s.t.} \ \ u_\kappa = \chi_{V_\kappa}, \ \forall\kappa \in \{1, \ldots, k\}. \tag{28}$$

As with classical spectral clustering, this problem is NP-hard due to the discrete constraints on $\mathbf{U}$ (Von Luxburg, 2007). To make it tractable, we adopt the standard relaxation: instead of binary indicator functions, we allow $\mathbf{U}$ to be any real-valued matrix with orthonormal columns. This leads to the relaxed optimization problem:

$$\min_{\mathbf{U}} \ \mathrm{tr}(\mathbf{U}^\mathsf{T}\mathbf{L}_\nu\mathbf{U}) \quad \text{s.t.} \ \ \mathbf{U}^\mathsf{T}\mathbf{U} = \mathbf{I}_k, \tag{29}$$

whose solution is given by the $k$ eigenvectors of $\mathbf{L}_\nu$ corresponding to its smallest eigenvalues.

The novelty of the proposed method lies in the generalized Laplacian $\mathbf{L}_\nu$, which is defined with respect to an arbitrary positive vertex measure $\nu$. This flexibility stems from the GDE framework and captures a rich family of partitioning behaviors grounded in random walk dynamics. When the measure is chosen as the stationary distribution, $\nu = \pi$, and the normalized Dirichlet energy (see Eq. 20) is minimized, the resulting method recovers the classical approach for strongly connected digraphs (Zhou et al., 2005).

***Clustering algorithm.*** The GSC framework relies on a parametrized vertex measure $\nu := \nu_{t,\alpha}$ introduced in Section 3.3. Given a pair of parameter values ($t \in \mathbb{N}$, $\alpha \in \mathbb{R}_+$), Eq. 23 defines the vertex measure $\nu_{t,\alpha}$ that induces a generalized Dirichlet energy $\mathcal{D}_{\nu_{t,\alpha}}^2(\boldsymbol{V})$ (Eq. 27). This, in turn, leads to the generalized spectral graph partitioning objective of Eq. 29, based on the generalized Laplacian $\mathbf{L}_{t,\alpha}$ (Eq. 16 or Eq. 18, depending on the variant). In practice, the GSC algorithm follows the same computational pipeline of the classical spectral clustering (see Algorithm 1), with the difference that it substitutes the standard Laplacian with the generalized one $\mathbf{L}_{t,\alpha}$, and computes its leading eigenvectors $\mathbf{U}_{t,\alpha} \in \mathbb{R}^{N \times k}$ for downstream clustering.

---

**Algorithm 1** Generalized Spectral Clustering (GSC)

---

**Input:** Weighted adjacency matrix $\mathbf{W}$; number of clusters $k$; diffusion time $t$; reweighting exponent $\alpha$.
**Output:** $\boldsymbol{V}_{t,\alpha}$: graph partition into $k$ clusters.

---

1: Compute the generalized Laplacian $\mathbf{L}_{t,\alpha}$ (Eq. 16).
2: Compute $\mathbf{U}_{t,\alpha} \in \mathbb{R}^{N \times k}$, whose columns are the eigenvectors corresponding to the $k$ smallest eigenvalues of $\mathbf{L}_{t,\alpha}$.
3: Embed each vertex $i$ into $\mathbb{R}^k$ using the $i$-th row of $\mathbf{U}_{t,\alpha}$, and apply a clustering algorithm to the embedded points.
4: Construct the vertex partition $\boldsymbol{V}_{t,\alpha} = \{V_{t,\alpha}^{(\kappa)}\}_{\kappa=1}^k$ based on clustering results.
5: **return** $\boldsymbol{V}_{t,\alpha}$

---

### 4.3 Cheeger-type inequalities for GDE minimization

In the context of spectral clustering, Cheeger-type inequalities provide a fundamental link between the optimal value of a clustering objective (e.g. the normalized cut) and the spectral properties of the associated graph Laplacian. These inequalities establish that if the second smallest eigenvalue of the Laplacian is small, then there exists a partition of the graph with low conductance, and conversely, if there exists a partition with low conductance, then the second smallest eigenvalue must also be small. This connection is crucial for understanding the theoretical guarantees of spectral clustering methods. In this subsection, we extend the Cheeger-type inequalities to the GDE minimization problem, providing theoretical insights into the quality of the partitions obtained by the GSC method. More specifically, for subset $S \subseteq \mathcal{V}$, we establish a conductance-like quantity defined with respect to the generalized edge measure $q(i,j) = \nu(i)p(i,j)$:

$$\phi_\nu(S) = \frac{q(S,\bar{S}) + q(\bar{S},S)}{\min\{\text{vol}_{\nu+\xi}(S), \text{vol}_{\nu+\xi}(\bar{S})\}} = \frac{\mathcal{D}_\nu^2(\chi_S)}{\min\{\text{vol}_{\nu+\xi}(S), \text{vol}_{\nu+\xi}(\bar{S})\}}. \tag{30}$$

Above, $\text{vol}_{\nu+\xi}(S) = \sum_{i \in S}(\nu(i)+\xi(i))$ is the volume of the set $S$ with respect to the vertex measure $\nu+\xi$, such that $\text{vol}_{\nu+\xi}(\mathcal{V}) = 2$. The quantity $\phi_\nu(S)$ measures the normalized bi-directional flow across the cut defined by $S$ and $\bar{S}$, where the normalization is done with respect to the vertex measure $\nu$. The bi-directional flow, $q(S,\bar{S}) + q(\bar{S},S)$, captures the total interaction between the two subsets, which is crucial for understanding the quality of a partition in terms of the dynamics induced by $\nu$.

Eq. 30 is a natural generalization of the classical conductance, which is defined with respect to the stationary distribution $\pi$ (Chung, 2005). Using Proposition 4.1 and its corollary, we see that when $\nu = \pi$, this formulation recovers the classical conductance, up to a multiplicative constant. This generalized conductance can be used to establish Cheeger-type inequalities that relate the optimal value of the GDE minimization problem to the quality of a partition in terms of $\phi_\nu$. We can thus define the vertex-measure-based Cheeger constant as:

$$\phi_\nu = \min_{S \subset \mathcal{V}} \phi_\nu(S),$$

which allows us to characterize the 'quality' of the best possible partition of the graph in terms of $\phi_\nu$.

**Theorem 4.1 (Cheeger-type inequalities for the GDE minimization problem).** *Let $\lambda_2$ be the second smallest eigenvalue of the normalized generalized Laplacian $\overline{\mathbf{L}}_\nu$. Then, the generalized conductance $\phi_\nu$ and $\lambda_2$ satisfy the following inequalities:*

$$\frac{\lambda_2}{2} \leq \phi_\nu \leq \sqrt{2\lambda_2}. \tag{31}$$

The proof is given in Section C.6 and relies on standard techniques from spectral graph theory, adapted to the generalized conductance. Note that using $\nu = \pi$ recovers the classical Cheeger inequalities for directed graphs (Chung, 2005), while using other vertex measures allows us to establish new Cheeger-type inequalities that are tailored to the specific dynamics induced by $\nu$. The lower bound indicates that if the second smallest eigenvalue ($\lambda_2$) is small, then there exists a partition of the graph with low generalized conductance $\phi_\nu$, which implies that the graph can be partitioned into two subsets with weak interaction under the dynamics induced by $\nu$. The upper bound indicates that if there exists a partition with low generalized conductance $\phi_\nu$, then the second smallest eigenvalue must also be small, which implies that the GDE minimization problem has a low optimal value. These inequalities provide theoretical guarantees for the performance of the GSC method regarding the partition quality it produces.

# 5 Experiments

This section presents an empirical evaluation of the proposed GSC method on clustering digraphs. The code implementing the methods and reproducing the experiments is available publicly[5].

## 5.1 Experimental setup

The proposed GSC method is empirically evaluated on clustering digraphs constructed from benchmark point-cloud data. The performance is compared against established spectral clustering baselines, under both fully unsupervised and label-informed evaluation protocols.

***Compared methods.*** We evaluate two GSC variants based on generalized Laplacians from Section 3.3:

- $\text{GSC}_{\text{un}}$ $(t, \alpha)$: using the unnormalized generalized Laplacian $\mathbf{L}_\nu$ (Eq. 16),
- $\text{GSC}_{\text{n}}$ $(t, \alpha)$: using the normalized generalized Laplacian $\overline{\mathbf{L}}_\nu$ (Eq. 18),

with $\nu := \nu_{t,\alpha}$ being the parametrized vertex measure (Eq. 23). For the UCI benchmark and the controlled synthetic clustering experiments, the grid search uses $t \in \{0, 1, ..., 25\}$ and $\alpha \in \{0.0, 0.1, ..., 1.5\}$. For the network benchmark, we use the same $t$ grid and extend the exponent grid to $\alpha \in \{0.0, 0.1, ..., 2.0\}$. The size-scaling runtime experiment uses a coarser grid only for timing. These upper limits are set by taking into account the scale of the graph sizes we deal with in the experiments. Additionally, a comparison of the proposed GSC variants with some natural alternatives to the $\nu_{t,\alpha}$ vertex measure is provided in Section B.

We compare against the following spectral clustering methods from the literature, the first four of which have been developed for directed graphs:

- DSC+$(\gamma)$ (Zhou et al., 2005; Palmer & Zheng, 2020), extended via teleporting random walks (Page et al., 1999) with teleportation parameter $\gamma$.
- DI-SIM$_\text{L}$, DI-SIM$_\text{R}$, and DI-SIM$_\text{C}$ (Rohe et al., 2016), which apply spectral clustering to the left, right, or concatenated singular vectors of a regularized transition matrix. We tune $s \in \{-1, -0.5, ..., 1\}$ and set the regularization parameter to $\tau(s) = \text{round}(\bar{d} \cdot 10^s)$, where $\bar{d}$ is the graph's average degree. The optimized $s$ is reported in the tables.
- $\text{SC}_{\text{un}}$ and $\text{SC}_{\text{n}}$: spectral clustering using unnormalized and normalized Laplacians on the symmetrized adjacency matrix (Von Luxburg, 2007).

All methods follow a standard spectral clustering pipeline: a graph embedding is computed via the eigenvectors of a Laplacian-type operator, followed by $k$-means**++** (100 restarts) clustering (Arthur & Vassilvitskii, 2007) on the rows of the embedding, where the number of clusters $k$ is assumed to be known. The best result is retained, according to an internal evaluation index.

***Clustering evaluation protocol.*** The clustering quality is assessed via the following metrics:

- *Internal evaluation*: We use an internal evaluation index, the Calinski-Harabasz index (CH), that computes the ratio between the between-cluster and within-cluster variance, both computed using cluster centers. Higher CH values are better. The CH index for a partition $\mathcal{C} = \{C_1, ..., C_k\}$ of $N$ datapoints is computed with respect to a distance or divergence function div:

$$\text{CH}(\mathcal{C}) = \frac{N - k}{k - 1} \cdot \frac{\sum_{j=1}^k |C_j| \operatorname{div}(c_j, c)}{\sum_{j=1}^k \sum_{i \in C_j} \operatorname{div}(x_i, c_j)}, \quad \text{where} \quad c_j = \frac{1}{|C_j|} \sum_{i \in C_j} x_i, \quad c = \frac{1}{N} \sum_{i=1}^N x_i.$$

  Above, div() is a divergence; for Euclidean point-cloud data, div() is the squared Euclidean distance. For network data, we represent each vertex $i$ by its outgoing one-step random-walk distribution $p_i = \delta_i^\mathsf{T} \mathbf{P}$ and apply the Hellinger embedding $z_i = \sqrt{p_i}$ before computing CH; we call this variant *Graph-CH* (GCH). In this geometry, $\|z_i - z_j\|_2^2 = 2(1 - \sum_k \sqrt{p_i(k) p_j(k)})$, so vertices are compared through probability overlap. This avoids applying raw Euclidean CH directly to probability rows, whose distances include variable

---

[5]Accessible online at: https://github.com/Malik-Hacini/Generalized-Spectral-Clustering

concentration terms $\|p_i\|_2^2 = \sum_k p_i(k)^2$, the self-collision probabilities of the random walks. Longer diffusion times $p_i = \delta_i^\mathsf{T} \mathbf{P}^t$, or mixtures of multiple diffusion times, can also be used to balance local and global information. In our experiments, we employ the standard CH for Euclidean data and the GCH for network data using the one-step random-walk representation $(t = 1)$.

- *External evaluation*: Adjusted Mutual Information (AMI) (Strehl & Ghosh, 2002) is an external evaluation measure computed post hoc and assesses the statistical agreement (adjusted for agreement by chance) between the found clusters and the ground-truth labels. Higher AMI values indicate better clustering.

A *fully unsupervised clustering evaluation protocol* is employed, which emulates real-world conditions where no data labeling is available during training or parameter selection:

- *Model selection via internal validation*: Each method explores its respective hyperparameter space: GSC sweeps over $(t, \alpha)$, DSC+ over $\gamma$, and DI-SIM over $s$, with $\tau(s)$ defined above. Each method performs 100 $k$-means++ runs per configuration of its hyperparameters, and the CH index, or GCH for network data, is computed for each clustering solution. By design, CH-type variance ratios prefer more 'regular' cluster shapes, meaning spherical (convex and compact) density shapes in the feature space used for validation. We present experimental results that focus on the clustering methods by mostly selecting datasets in which this internal-index choice is suitable. A workaround to this can be to make a more informed choice of the internal evaluation index and use it with the same evaluation pipeline, or to go further and find ways to combine multiple evaluation indices.

- *Evaluation and reporting*: The solution yielding the best internal score (CH, or GCH for network data), measured with respect to the original data representation space, is retained along with the associated optimal hyperparameters; e.g. for GSC, those would be $(t, \alpha)^*_{\mathrm{CH}} := (t^*_{\mathrm{CH}}, \alpha^*_{\mathrm{CH}})$. The AMI of that best solution is reported post hoc to better interpret the results using an external evaluation index.

The above protocol is designed to be fully unsupervised; the key point is that the model selection process relies solely on an internal validation index, without any access to ground-truth labels. The choice of the internal evaluation index influences the results. In all cases, we employ the generic and well-understood CH index (and the GCH variant for network data); however, this is not a hardwired feature of the proposed framework or pipeline. Thus, a user might consider selecting an internal index that aligns well with the expected cluster structure of the data. As a sanity check, we also report the best AMI obtained by each method when the hyperparameters are optimized directly on the AMI, which is a supervised evaluation protocol that serves as an 'upper bound' for the unsupervised results. As a summary, we report the *Performance Ratio to Best* (PRB) of each method, defined as the mean of the ratios between a quality metric $\mathcal{Q}$ of each method and the best such index across all methods, for each dataset. PRB quantifies how close a method is to the best performing method in each dataset, with values closer to 1 indicating better performance:

$$\mathrm{PRB}_{\mathcal{Q}}(m) = \frac{1}{|\mathcal{D}|} \sum_{j \in \mathcal{D}} \frac{\mathcal{Q}_j(m)}{\max_{m'} \mathcal{Q}_j(m')},$$

where $\mathcal{D}$ is the collection of all datasets, $\mathcal{Q}_j(m)$ is the score of the method $m$ on dataset $j$ according to the metric $\mathcal{Q}$, and $\max_{m'} \mathcal{Q}_j(m')$ is the maximum score achieved by any method on dataset $j$.

***Graph construction from point-clouds.*** We test the clustering performance across different real-world datasets from the UCI repository[6]. Given a dataset $\boldsymbol{X} = \{x_i\}_{i=1}^N$ with $x_i \in \mathbb{R}^d$, we build a sparse directed graph using an unweighted, directed $K$-nearest neighbor ($K$-NN) construction, where $K = \max\{1, \lfloor \log(N) \rfloor\}$. This process yields sparse and typically weakly connected digraphs. The unweighted and asymmetric adjacency matrix $\mathbf{W} = \{w_{ij}\}$ is defined as: $w_{ij} = \mathbb{1}\{\|x_i - x_j\|^2 \le \mathrm{dist}_K(x_i)^2\}$, where $\mathrm{dist}_K(x_i)$ denotes the distance of $x_i$ to its $K$-th nearest neighbor.

***Network data.*** We evaluate the clustering performance also on real-world networks, including PolBlogs, Email-Eu-Core, PolBooks, College Football, a synthetic directed stochastic block model (DiSBM Chain) with 3 communities of 500 vertices each, organized in a chain structure, and a Degree-Corrected DiSBM (Deg-Corr), where one community has a higher average degree than the others. For these datasets, we use

---

[6]Accessible online at: https://archive.ics.uci.edu/

the original adjacency matrices as input to the clustering methods, without any additional preprocessing or symmetrization.

## 5.2 Results

This section presents extensive empirical results following the described protocol. We start with clustering results on a number of benchmark point-cloud and network datasets, for which we also perform a sensitivity analysis over the parametrization of the proposed vertex measure, and report runtime results confirming that the computational cost of GSC remains reasonable. Then, we explore three challenging scenarios, using a controlled protocol and synthetic data. Those scenarios concern the impact of directionality in clustering through the usage of a chain-structured DiSBM as well as clustering under cluster size-imbalance and degree imbalance (inhomogeneity). Although dealing with these scenarios is not among the main aims of the present work, they constitute characteristic challenges in clustering for which we demonstrate that GSC has a more flexible and robust behavior compared to the other methods. Finally, we summarize the results in the last part of the section.

### 5.2.1 Experiments on benchmark datasets

***Clustering performance.*** Table 1 and Table 2 present clustering results across UCI datasets and networks. Each table consists of three subtables: the first refers to the unsupervised optimization of each method with respect to the CH (or GCH) criterion, for which the precise objective values are reported; the second subtable contains the AMI scores corresponding to the same clustering results of the first subtable; the third subtable shows the clustering results for when the methods are optimized with respect to ground-truth labels. As explained in Section 5.1, the latter results are provided only as a reference to demonstrate the capacity of each model without the effect of the chosen internal validation index.

*Supervised results.* Under direct AMI-based tuning, both GSC variants remain strong. On UCI datasets, $GSC_{un}$ and $GSC_n$ achieve $PRB$(AMI) scores of 0.94 and 0.99, respectively, with $GSC_n$ attaining the top performance on 7 of the 9 datasets. On networks, their $PRB$(AMI) scores are 0.84 and 0.89. The normalized variant is therefore close to the strongest performance, attained by $DI\text{-}SIM_C$ with a $PRB$(AMI) of 0.91. On networks, $SC_n$ remains moderately competitive with a $PRB$(AMI) of 0.73, whereas $SC_{un}$ drops to 0.46, with zero AMI on PolBlogs, Deg-Corr, and Email-Eu-Core. DSC+ obtains 0.76, while $DI\text{-}SIM_R$, $DI\text{-}SIM_L$, and $DI\text{-}SIM_C$ obtain 0.87, 0.89, and 0.91, respectively.

*Unsupervised results.* Under CH/GCH-based tuning, both GSC variants remain strong. On UCI datasets, $GSC_{un}$ and $GSC_n$ achieve $PRB$(CH) scores of 0.96 and 0.99, respectively. On networks, their $PRB$(GCH) scores are 0.83 and 0.86, while their corresponding $PRB$(AMI) scores are 0.83 and 0.89. The normalized variant is therefore close to the strongest external-label performance, attained by $DI\text{-}SIM_C$ with a $PRB$(AMI) of 0.91. On networks, $SC_n$ remains moderately competitive with a $PRB$(AMI) of 0.73, whereas $SC_{un}$ drops to 0.46, with zero AMI on PolBlogs, Deg-Corr, and Email-Eu-Core. DSC+ obtains 0.75, while $DI\text{-}SIM_R$, $DI\text{-}SIM_L$, and $DI\text{-}SIM_C$ obtain 0.81, 0.88, and 0.91, respectively.

On networks, GCH-based tuning recovers the same $GSC_n$ hyperparameters and AMI scores as direct AMI tuning on five of the six datasets, with Email-Eu-Core being the exception. For $GSC_{un}$, the AMI is also unchanged except on PolBooks and Email-Eu-Core; on DiSBM Chain, different selected hyperparameters yield the same AMI of 0.73. Thus, the network results do not indicate greater sensitivity of the normalized formulation to unsupervised tuning.

*Remark on the top-ranking methods.* On the network datasets, the two leading methods are close: $DI\text{-}SIM_C$ attains a $PRB$(AMI) of 0.91 and $GSC_n$ of 0.89. We caution against reading this ordering as definitive: during our experimentation, we observed that the relative ranking of GSC and DI-SIM on real network data is sensitive to specific choices of the experimental pipeline, notably the construction of the reference representation on which GCH operates (e.g. the diffusion time and the embedding of the random-walk profiles), the divergence used to compare vertices, and the tuning ranges of the methods' regularization parameters. The reported results follow a single protocol applied uniformly to all methods and scenarios, and thus offer a fair common ground; under reasonable variations of these components, however, the two top

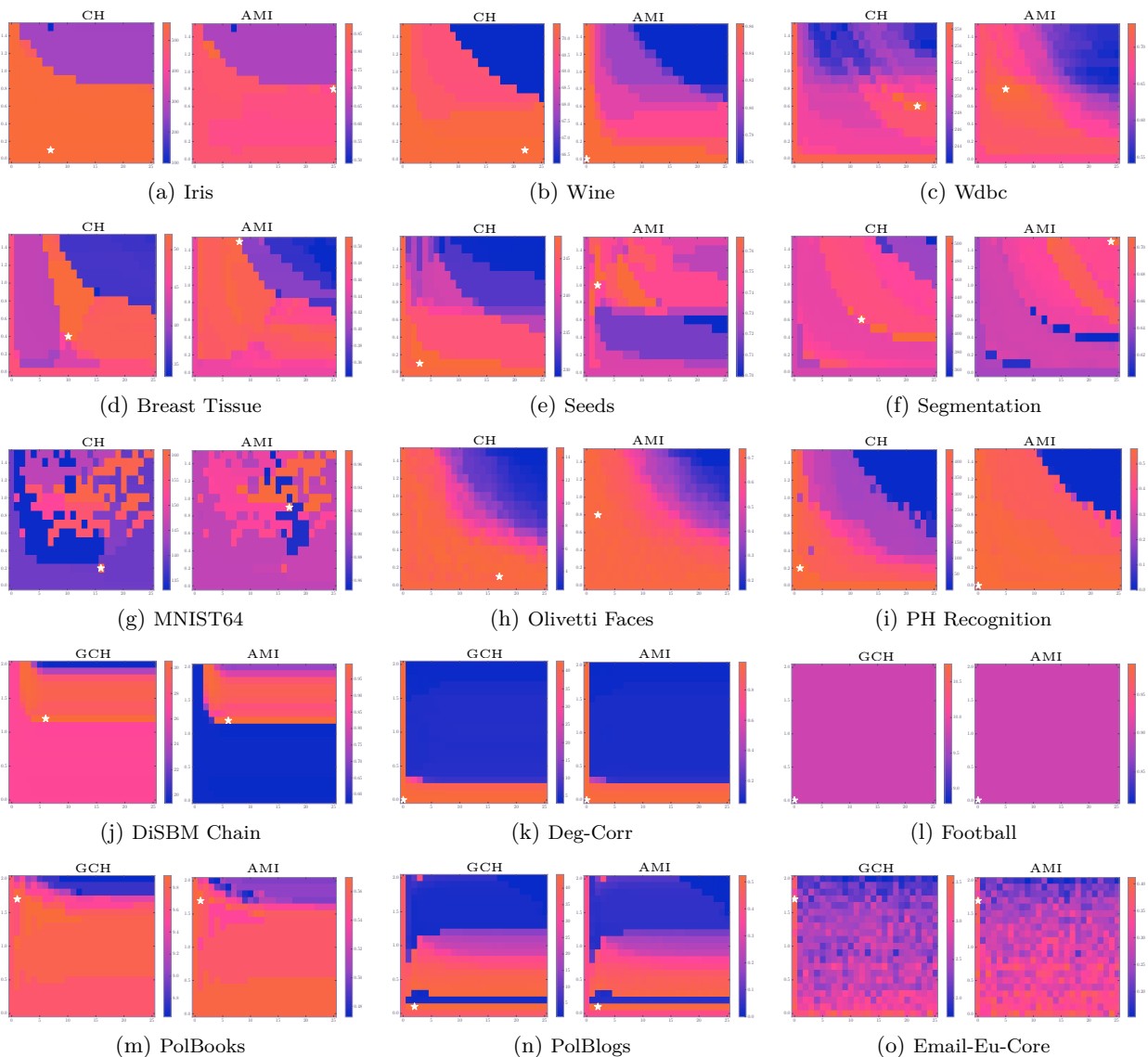

Figure 2: **GSC$_n$ sensitivity to hyperparameters.** For each dataset, CH/GCH and AMI heatmaps are shown over the space of vertex measure parametrization ($t$: x-axis, $\alpha$: y-axis). Shades of orange indicate better clustering, and a white star marks the best configuration for each index, i.e. $(t, \alpha)^*_{\text{CH}}$ vs $(t, \alpha)^*_{\text{AMI}}$. In most cases, the CH/GCH landscape aligns well with that of AMI, supporting internal-index-based unsupervised model selection. (a-i) Heatmaps for the UCI point-cloud datasets; (j-o) heatmaps for the network datasets. For Wdbc, Seeds and Segmentation, the CH and AMI landscapes differ, highlighting the difficulty of unsupervised model selection when the internal index does not align well with the cluster structure of the data. In datasets like MNIST64, the behavior of both CH and AMI exhibits high sensitivity to the choice of vertex measure, with very scattered regions. The correspondence between the two still appears relevant in this context.

methods can swap order, while the overall qualitative picture remains unchanged. We therefore regard GSC$_n$ and DI-SIM$_C$ as of essentially equivalent performance in this regime.

***Sensitivity analysis for the parametrized vertex measure.*** Here, we analyze how the GSC clustering quality depends on the vertex measure, and we validate that an internal index such as CH, or GCH for network data, can be used as a surrogate index for unsupervised model selection, through hyperparameter tuning. Fig. 2 illustrates the sensitivity of the vertex measure $\nu_{t,\alpha}$ (Eq. 23) to its hyperparameters. For

Table 1: **Comparison of clustering methods on UCI datasets.** (a-b) Unsupervised evaluation by optimizing the CH criterion: the first table shows CH scores, while the second shows the corresponding AMI scores. (c) Supervised evaluation by optimizing AMI directly. Optimized hyperparameters are shown in parentheses. Bold on shaded background marks the best-performing method per dataset, based on unrounded scores.

(a) UNSUPERVISED EVALUATION (CH SCORES | CH-OPTIMIZED)

| Dataset | $SC_{un}$ | $SC_n$ | $DSC+(\gamma)$ | $DI\text{-}SIM_R(s)$ | $DI\text{-}SIM_L(s)$ | $DI\text{-}SIM_C(s)$ | $GSC_{un}(t,\alpha)$ | $GSC_n(t,\alpha)$ |
|---|---|---|---|---|---|---|---|---|
| Breast Tissue | 50.98 | 47.79 | 26.66 (0.05) | 54.22 (-1.00) | 54.41 (-0.50) | 54.27 (-0.50) | **54.78** (1, 0.4) | 52.04 (10, 0.4) |
| Iris | 555.67 | 555.67 | 22.71 (0.05) | 501.63 (-1.00) | 496.70 (-1.00) | 501.63 (-1.00) | 555.67 (0, 0.0) | **558.06** (7, 0.1) |
| MNIST64 | 139.69 | 139.69 | 52.52 (0.15) | 130.05 (-0.50) | 159.46 (-0.50) | 158.47 (-0.50) | 139.69 (0, 0.0) | **161.54** (3, 0.5) |
| Olivetti Faces | 14.42 | 14.40 | 9.92 (0.15) | 10.49 (-0.50) | 13.63 (0.00) | 12.71 (-0.50) | 14.42 (0, 0.0) | **14.88** (17, 0.1) |
| PH Recognition | 436.85 | 436.85 | 9.50 (0.05) | 24.85 (-1.00) | 46.45 (-1.00) | 44.65 (-1.00) | 436.85 (0, 0.0) | **438.12** (2, 0.1) |
| Seeds | 247.31 | 247.31 | 151.03 (0.20) | 230.54 (-1.00) | 237.09 (-1.00) | 231.10 (-1.00) | 247.31 (0, 0.0) | **248.04** (3, 0.1) |
| Segmentation | 419.00 | 435.75 | 10.52 (0.50) | 431.46 (-1.00) | 467.61 (-0.50) | 467.06 (-0.50) | 419.00 (0, 0.0) | **507.94** (12, 0.6) |
| WDBC | 257.69 | 257.69 | 102.32 (0.10) | 219.76 (-1.00) | 257.29 (-1.00) | 256.08 (-1.00) | 257.69 (0, 0.0) | **258.74** (22, 0.6) |
| Wine | 70.37 | 70.37 | 70.43 (0.05) | 69.39 (-1.00) | 69.82 (1.00) | 70.29 (0.50) | 70.37 (0, 0.0) | **70.44** (22, 0.1) |
| $PRB$(CH) | 0.95 | 0.95 | 0.40 | 0.79 | 0.86 | 0.85 | 0.96 | **0.99** |

(b) UNSUPERVISED EVALUATION (AMI SCORES | CH-OPTIMIZED)

| Dataset | $SC_{un}$ | $SC_n$ | $DSC+(\gamma)$ | $DI\text{-}SIM_R(s)$ | $DI\text{-}SIM_L(s)$ | $DI\text{-}SIM_C(s)$ | $GSC_{un}(t,\alpha)$ | $GSC_n(t,\alpha)$ |
|---|---|---|---|---|---|---|---|---|
| Breast Tissue | 0.46 | 0.45 | 0.28 (0.05) | 0.46 (-1.00) | **0.50** (-0.50) | 0.50 (-0.50) | 0.48 (1, 0.4) | 0.48 (10, 0.4) |
| Iris | **0.80** | **0.80** | 0.35 (0.05) | 0.75 (-1.00) | 0.75 (-1.00) | 0.75 (-1.00) | **0.80** (0, 0.0) | 0.77 (7, 0.1) |
| MNIST64 | 0.91 | 0.91 | 0.42 (0.15) | 0.81 (-0.50) | 0.97 (-0.50) | **0.97** (-0.50) | 0.91 (0, 0.0) | 0.95 (3, 0.5) |
| Olivetti Faces | 0.70 | **0.72** | 0.63 (0.15) | 0.63 (-0.50) | 0.71 (0.00) | 0.70 (-0.50) | 0.70 (0, 0.0) | 0.70 (17, 0.1) |
| PH Recognition | **0.56** | **0.56** | 0.18 (0.05) | 0.25 (-1.00) | 0.34 (-1.00) | 0.34 (-1.00) | **0.56** (0, 0.0) | 0.56 (2, 0.1) |
| Seeds | **0.74** | **0.74** | 0.61 (0.20) | 0.71 (-1.00) | 0.68 (-1.00) | 0.70 (-1.00) | **0.74** (0, 0.0) | 0.74 (3, 0.1) |
| Segmentation | 0.64 | **0.64** | 0.06 (0.50) | 0.63 (-1.00) | 0.63 (-0.50) | 0.62 (-0.50) | 0.64 (0, 0.0) | 0.61 (12, 0.6) |
| WDBC | **0.68** | **0.68** | 0.46 (0.10) | 0.64 (-1.00) | 0.66 (-1.00) | **0.68** (-1.00) | 0.68 (0, 0.0) | 0.62 (22, 0.6) |
| Wine | 0.86 | 0.86 | 0.86 (0.05) | **0.86** (-1.00) | 0.83 (1.00) | 0.84 (0.50) | 0.86 (0, 0.0) | 0.85 (22, 0.1) |
| $PRB$(AMI) | 0.98 | 0.98 | 0.58 | 0.88 | 0.93 | 0.93 | **0.98** | 0.97 |

(c) SUPERVISED EVALUATION (AMI SCORES | AMI-OPTIMIZED)

| Dataset | $SC_{un}$ | $SC_n$ | $DSC+(\gamma)$ | $DI\text{-}SIM_R(s)$ | $DI\text{-}SIM_L(s)$ | $DI\text{-}SIM_C(s)$ | $GSC_{un}(t,\alpha)$ | $GSC_n(t,\alpha)$ |
|---|---|---|---|---|---|---|---|---|
| Breast Tissue | 0.46 | 0.45 | 0.29 (0.15) | 0.47 (-0.50) | 0.50 (-0.50) | 0.50 (-0.50) | 0.51 (21, 0.1) | **0.51** (8, 1.5) |
| Iris | 0.80 | 0.80 | 0.35 (0.05) | 0.76 (-0.50) | 0.75 (-1.00) | 0.75 (-1.00) | 0.80 (0, 0.0) | **0.88** (25, 0.8) |
| MNIST64 | 0.91 | 0.91 | 0.42 (0.80) | 0.86 (-1.00) | 0.97 (-0.50) | **0.97** (-0.50) | 0.91 (0, 0.0) | 0.97 (17, 0.9) |
| Olivetti Faces | 0.70 | 0.72 | 0.63 (0.15) | 0.63 (-0.50) | 0.71 (0.00) | 0.70 (-0.50) | 0.70 (0, 0.0) | **0.74** (2, 0.8) |
| PH Recognition | **0.56** | **0.56** | 0.18 (0.05) | 0.25 (-1.00) | 0.34 (-1.00) | 0.34 (-1.00) | **0.56** (0, 0.0) | **0.56** (0, 0.0) |
| Seeds | 0.74 | 0.74 | 0.61 (0.20) | 0.71 (-1.00) | 0.69 (-0.50) | 0.70 (-1.00) | 0.74 (0, 0.0) | **0.77** (2, 1.0) |
| Segmentation | 0.64 | 0.64 | 0.07 (0.95) | 0.63 (-1.00) | 0.64 (-1.00) | 0.64 (-1.00) | 0.64 (0, 0.0) | **0.71** (24, 1.5) |
| WDBC | 0.68 | 0.68 | 0.46 (0.10) | 0.64 (-1.00) | 0.69 (-0.50) | 0.69 (-0.50) | 0.68 (0, 0.0) | **0.73** (5, 0.8) |
| Wine | 0.86 | 0.86 | 0.86 (0.05) | **0.95** (0.00) | 0.83 (1.00) | 0.87 (1.00) | 0.86 (0, 0.0) | 0.86 (0, 0.0) |
| $PRB$(AMI) | 0.93 | 0.93 | 0.55 | 0.85 | 0.89 | 0.89 | 0.94 | **0.99** |

Table 2: **Comparison of clustering methods on network datasets.** (a-b) Unsupervised evaluation by optimizing GCH on Hellinger-embedded one-step random-walk profiles: the first table shows GCH scores, while the second shows the corresponding AMI scores. (c) Supervised evaluation by optimizing AMI directly. Optimized hyperparameters are shown in parentheses. Bold on shaded background marks the best-performing method per dataset, based on unrounded scores.

(a) UNSUPERVISED EVALUATION (GCH SCORES | GCH-OPTIMIZED)

| Dataset | $SC_{un}$ | $SC_{n}$ | $DSC+(\gamma)$ | $DI\text{-}SIM_{R}(s)$ | $DI\text{-}SIM_{L}(s)$ | $DI\text{-}SIM_{C}(s)$ | $GSC_{un}(t,\alpha)$ | $GSC_{n}(t,\alpha)$ |
|---|---|---|---|---|---|---|---|---|
| Deg-Corr | 1.09 | 43.03 | **43.35** (0.70) | 43.28 (0.50) | **43.35** (-1.00) | **43.35** (-1.00) | 42.85 (0, 0.0) | 42.76 (0, 0.0) |
| DiSBM Chain | 13.90 | 25.85 | 14.26 (0.20) | 29.57 (-1.00) | 30.46 (-1.00) | **30.57** (-1.00) | 26.45 (3, 0.2) | 30.56 (6, 1.2) |
| Email-Eu-Core | 0.88 | 4.53 | 1.26 (0.10) | 7.64 (0.00) | **8.49** (-0.50) | 8.40 (-0.50) | 3.37 (0, 0.0) | 3.58 (23, 0.0) |
| Football | **9.77** | **9.77** | 9.77 (0.05) | 9.77 (-1.00) | 9.77 (-1.00) | 9.77 (-1.00) | 9.77 (0, 0.0) | 9.77 (0, 0.0) |
| PolBlogs | 1.18 | 1.18 | **59.08** (0.10) | 28.14 (-1.00) | 51.99 (-1.00) | 49.98 (-1.00) | 44.98 (0, 0.0) | 44.19 (2, 0.1) |
| PolBooks | 7.69 | 9.69 | 9.69 (0.05) | **10.00** (-0.50) | **10.00** (-0.50) | **10.00** (-0.50) | 9.73 (1, 0.3) | 9.91 (1, 1.7) |
| $PRB$(GCH) | 0.40 | 0.73 | 0.76 | 0.89 | **0.98** | 0.97 | 0.83 | 0.86 |

(b) UNSUPERVISED EVALUATION (AMI SCORES | GCH-OPTIMIZED)

| Dataset | $SC_{un}$ | $SC_{n}$ | $DSC+(\gamma)$ | $DI\text{-}SIM_{R}(s)$ | $DI\text{-}SIM_{L}(s)$ | $DI\text{-}SIM_{C}(s)$ | $GSC_{un}(t,\alpha)$ | $GSC_{n}(t,\alpha)$ |
|---|---|---|---|---|---|---|---|---|
| Deg-Corr | 0.00 | 0.99 | **1.00** (0.70) | 0.69 (0.50) | 0.98 (-1.00) | **1.00** (-1.00) | 0.99 (0, 0.0) | 0.98 (0, 0.0) |
| DiSBM Chain | 0.73 | 0.53 | 0.65 (0.20) | 0.96 (-1.00) | 0.97 (-1.00) | **1.00** (-1.00) | 0.73 (3, 0.2) | **1.00** (6, 1.2) |
| Email-Eu-Core | 0.00 | **0.51** | 0.00 (0.10) | 0.49 (0.00) | 0.46 (-0.50) | 0.50 (-0.50) | 0.37 (0, 0.0) | 0.39 (23, 0.0) |
| Football | **0.90** | **0.90** | 0.90 (0.05) | 0.90 (-1.00) | 0.90 (-1.00) | 0.90 (-1.00) | 0.90 (0, 0.0) | 0.90 (0, 0.0) |
| PolBlogs | 0.00 | 0.00 | **0.77** (0.10) | 0.30 (-1.00) | 0.47 (-1.00) | 0.47 (-1.00) | 0.54 (0, 0.0) | 0.52 (2, 0.1) |
| PolBooks | **0.64** | 0.57 | 0.57 (0.05) | 0.53 (-0.50) | 0.53 (-0.50) | 0.53 (-0.50) | 0.55 (1, 0.3) | 0.57 (1, 1.7) |
| $PRB$(AMI) | 0.46 | 0.73 | 0.75 | 0.81 | 0.88 | **0.91** | 0.83 | 0.89 |

(c) SUPERVISED EVALUATION (AMI SCORES | AMI-OPTIMIZED)

| Dataset | $SC_{un}$ | $SC_{n}$ | $DSC+(\gamma)$ | $DI\text{-}SIM_{R}(s)$ | $DI\text{-}SIM_{L}(s)$ | $DI\text{-}SIM_{C}(s)$ | $GSC_{un}(t,\alpha)$ | $GSC_{n}(t,\alpha)$ |
|---|---|---|---|---|---|---|---|---|
| Deg-Corr | 0.00 | 0.99 | **1.00** (0.70) | 0.97 (-0.50) | 0.98 (-1.00) | **1.00** (-1.00) | 0.99 (0, 0.0) | 0.98 (0, 0.0) |
| DiSBM Chain | 0.73 | 0.53 | 0.65 (0.90) | 0.96 (-1.00) | 0.97 (-1.00) | **1.00** (-1.00) | 0.73 (0, 0.0) | **1.00** (6, 1.2) |
| Email-Eu-Core | 0.00 | 0.51 | 0.00 (0.00) | **0.53** (-1.00) | 0.46 (-0.50) | 0.50 (-1.00) | 0.38 (25, 0.1) | 0.41 (0, 1.7) |
| Football | **0.90** | **0.90** | 0.90 (0.05) | 0.90 (-1.00) | 0.90 (-1.00) | 0.90 (-1.00) | 0.90 (0, 0.0) | 0.90 (0, 0.0) |
| PolBlogs | 0.00 | 0.00 | **0.77** (0.10) | 0.30 (-1.00) | 0.47 (-1.00) | 0.47 (-1.00) | 0.54 (0, 0.0) | 0.52 (2, 0.1) |
| PolBooks | **0.64** | 0.57 | 0.57 (0.05) | 0.57 (-1.00) | 0.57 (-1.00) | 0.57 (-1.00) | 0.58 (8, 0.3) | 0.57 (1, 1.7) |
| $PRB$(AMI) | 0.46 | 0.73 | 0.76 | 0.87 | 0.89 | **0.91** | 0.84 | 0.89 |

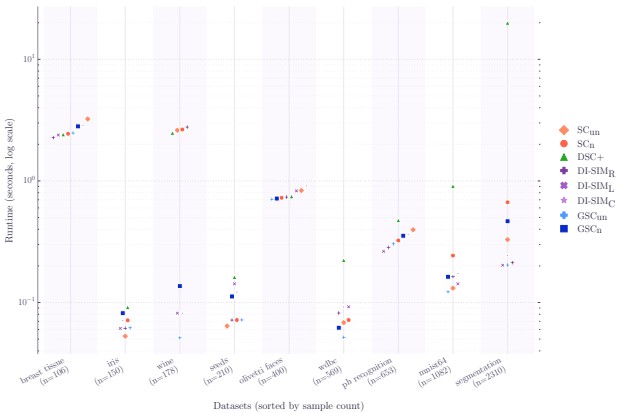

(a) Runtime of the algorithms across the UCI datasets.

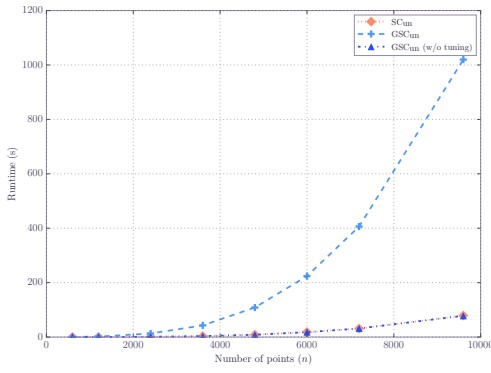

(b) Runtime of GSC$_{un}$ with and without hyper-parameter tuning, compared to SC$_{un}$.

Figure 3: **Runtimes.** (a) Runtimes for all methods across the UCI datasets. (b) Runtimes of GSC$_{un}$ with and without hyperparameter tuning, compared to SC$_{un}$, as the number of points increases. The results show that GSC is computationally as efficient as vanilla SC, while hyperparameter tuning brings reasonable overhead. The unnormalized GSC and SC variants are generally faster than their normalized counterparts.

each dataset, two heatmaps are shown with the landscapes produced for the internal index and the AMI index. Shades of orange indicate clustering results of higher quality. A white star in each plot indicates the hyperparameter configuration that yields the best respective evaluation index. Although some variability arises from the $k$-means++ initialization, for those datasets poor-quality regions can be identified in the CH/GCH landscape, which aligns well with the AMI landscape in high-quality regions. For datasets with well-separated clusters that are easier to detect, the clustering quality is relatively insensitive to the choice of $\nu_{t,\alpha}$; often the measure relies on $\alpha$ or $t$ being equal to 0. Conversely, in more challenging cases, the choice of measure becomes crucial, and identifying the best hyperparameter becomes a task of interest. These results support using a suitable internal index as a reliable proxy for guiding model selection in fully unsupervised settings.

***Runtimes.*** The computational efficiency of GSC is evaluated by measuring the runtime of the clustering pipeline across datasets of varying sizes. In particular, we show both the runtime of the different algorithms across the UCI datasets, and the runtime of GSC$_{un}$ with and without hyperparameter tuning, compared to the runtime of SC$_{un}$ in context where the number of points varies. The results (Fig. 3) indicate that GSC$_{un}$ without tuning has a runtime close to that of SC$_{un}$, demonstrating that the core algorithmic steps are as scalable as standard spectral clustering. Indeed, the main computational steps of GSC involve the construction of the graph Laplacian and the eigen-decomposition, which are similar to those in vanilla spectral clustering. The additional overhead from computing the vertex measure $\nu_{t,\alpha}$ is negligible compared to these steps, as it can be computed efficiently using power iterations or closed-form expressions for certain choices of $t$ and $\alpha$. The main overhead arises from the tuning of hyperparameters $(t, \alpha)$, which involves multiple runs of the clustering pipeline. However, even with tuning, the runtime remains reasonable in the benchmark datasets. In larger scale datasets, the tuning overhead can be mitigated by employing more efficient search strategies (e.g., random search, Bayesian optimization), by leveraging parallel computing resources or by using a non-tunable vertex measure (Section B). Note that DSC+ is generally quite close to GSC in terms of runtimes, while having a smaller grid-search size; this is due to the teleportation random walk, which implies a dense matrix, making the eigen-decomposition step more expensive. The unnormalized GSC and SC variants are generally faster than their normalized counterparts.

### 5.2.2 Exploration of challenging synthetic scenarios

We investigate three synthetic scenarios that are challenging for vanilla SC, and therefore informative for assessing the benefits of GSC. Following the intuition in Section 3.1, we focus on cases where skewed stationary

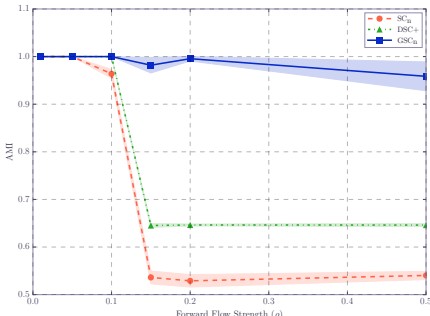

Figure 4: **Clustering performance in the chain-structured digraph setting.** $GSC_n$, $SC_n$ and DSC+ are evaluated on a synthetic chain-structured directed graph. We vary the forward edge probability $\rho$ and measure the AMI score of the clustering solutions.

distributions hinder the recovery of the cluster structure by SC. We compare GSC with vanilla SC and DSC+, a strong directed-graph baseline based on teleportation random walks.

***Chain-structured digraphs.*** In this setting, the graph is composed of 3 clusters of 500 points, arranged in a chain structure, with forward edges from cluster 1 to cluster 2, and from cluster 2 to cluster 3 (see details in Section A). The forward edge probability $\rho$ controls the strength of the directed structure, with higher $\rho$ leading to more pronounced directionality. The backward edge probability is fixed to 0.01. This yields a setting where the stationary distribution of the natural random walk is skewed, with most of the mass concentrated in the last cluster. The results in Fig. 4 show that GSC significantly outperforms SC and DSC+ across all values of $\rho$. In particular, as $\rho$ increases, the performance of both SC and DSC+ degrades significantly, while GSC remains robust and maintains a high AMI score. This demonstrates that GSC is able to effectively leverage the directed structure of the graph to recover meaningful clusters, even in cases where $SC_n$ fails due to skewed stationary distributions.

***Cluster size-imbalance.*** As highlighted in Section 3.1, the case of skewed stationary distribution is of particular interest, as it can lead to poor clustering for classical SC methods. To investigate this, we construct synthetic datasets with varying size imbalance, and evaluate the performance of GSC against $SC_n$ and DSC+. The datasets are constructed as follows: we place $K$ clusters on a squared grid, with alternating sizes. The smaller clusters contain $n_{low}$ points, while the larger clusters contain $n_{high} = 300$ points. Datapoints are sampled from Gaussian distributions centered at the cluster centroids, with a fixed variance $\sigma^2 = 0.3$, and cluster centroids are spaced out by 3. We vary the number of clusters $K$ and the size of the smaller clusters $n_{low}$ to create different levels of imbalance. The results (Fig. 5) show that GSC is more effective at recovering small clusters compared to vanilla SC, which tends to merge them into larger clusters. In particular, as the number of clusters grows, we can see that the ratio needed to obtain full performance becomes larger. In every case, GSC recovers the structure for smaller ratios of density than vanilla SC. In all considered cases, DSC+ performs poorly, because the stationary distribution is very skewed and because it relies on the teleportation random walk, which may obscure the information contained in the original graph. While not shown in the figures, DI-SIM variants perform similarly to $SC_n$.

***Vertex degree-imbalance.*** In degree-imbalanced graphs, the stationary distribution of the natural random walk is skewed, which can lead to poor clustering for classical SC methods. To investigate this, we construct synthetic directed graphs with varying degrees of degree imbalance and evaluate the performance of GSC against $SC_n$ and DSC+. The graphs are constructed as follows: starting from 3 Gaussian clusters of 300 points, we construct a "Gaussian injection" using:

$$\mathbf{K}(i,j) = \exp\left(-\frac{\|x_i - x_j\|^2}{2s^2}\right) \quad \text{and} \quad \tilde{\mathbf{K}}(i,j) = \exp\left(-\frac{\|x_i - c\|^2 + \|x_j - c\|^2}{2\sigma^2}\right),$$

where $c$ is the fixed center of the Gaussian injection (here, the center of the first cluster). The parameter $\sigma$ controls the level of degree imbalance, with smaller $\sigma$ leading to more skewed degree distributions. We then construct the adjacency matrix $\mathbf{A}_\alpha$, by combining the base kernel $\mathbf{K}$ and the bias kernel $\tilde{\mathbf{K}}$, as

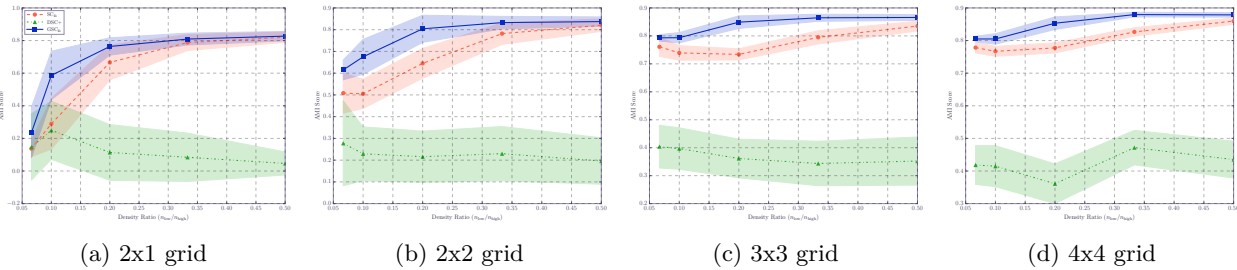

| (a) 2x1 grid | (b) 2x2 grid | (c) 3x3 grid | (d) 4x4 grid |

Figure 5: **Clustering performance in the size-imbalance setting.** $\text{GSC}_\text{n}$, $\text{SC}_\text{n}$ and DSC+ are evaluated on synthetic datasets with varying levels of cluster size imbalance. The x-axis represents the ratio of points in smaller clusters to larger clusters, while the y-axis shows the AMI score.

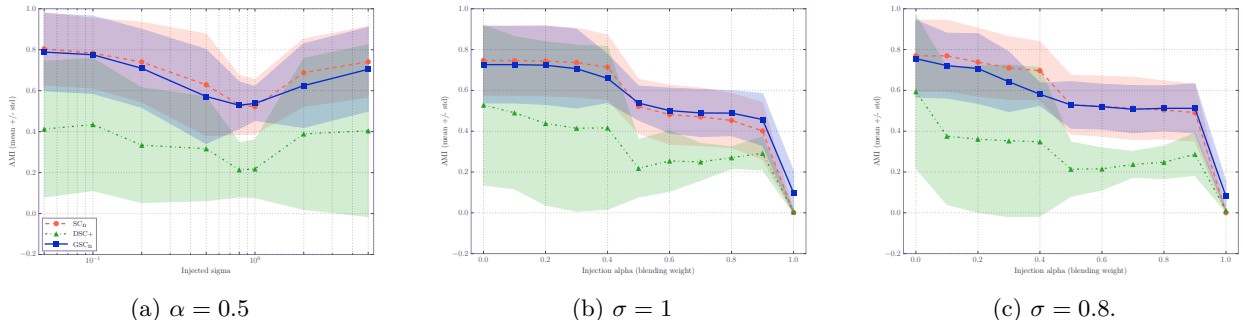

| (a) $\alpha = 0.5$ | (b) $\sigma = 1$ | (c) $\sigma = 0.8$. |

Figure 6: **Gaussian noise injection.** $\text{GSC}_\text{n}$, $\text{SC}_\text{n}$ and DSC+ are evaluated on synthetic directed graphs with varying levels of degree imbalance. In (a), the x-axis represents the standard deviation $\sigma$ of the Gaussian noise added to the edge weights; in (b-c), the x-axis represents the hyperparameter $\alpha$. The y-axis shows the mean AMI score obtained by the methods, over multiple seeds. The results indicate that $\text{GSC}_\text{n}$ and $\text{SC}_\text{n}$ outperform DSC+, which is not able to recover the cluster structure due to the skewed stationary distribution.

$\mathbf{A}_\alpha = (1 - \alpha)\mathbf{K} + \alpha\tilde{\mathbf{K}}$. A $K$-NN graph is then constructed from $\mathbf{A}_\alpha$, with $K = \lfloor \log(N) \rfloor$. The parameter $\alpha$ controls the strength of the degree imbalance, with larger $\alpha$ leading to more skewed degree distributions. Results are shown in Fig. 6. In Fig. 6a, we investigate, for fixed $\alpha = 0.5$, how the different methods perform for different $\sigma$ values. In Figs. 6b and 6c, we investigate, for $\sigma$ fixed, how the different methods perform for different $\alpha$ values. In all cases, $\text{GSC}_\text{n}$ and $\text{SC}_\text{n}$ outperform DSC+, which is not able to recover the cluster structure due to the skewed stationary distribution.

### 5.2.3 Summary of the results

As a whole, the reported experiments highlight the flexibility and robustness of the GSC framework. The tunable parameters $(t, \alpha)$ enable adaptation to diverse graph topologies, from sparse and directed to dense and symmetric. The observed alignment between internal (CH/GCH) and external (AMI) metrics across a variety of datasets supports the practical value of internal-index-based model selection in fully unsupervised settings, provided that the internal index is suitable for the cluster structure of a dataset. By generalizing Dirichlet energy through flexible vertex measures, GSC provides a principled and scalable approach to graph clustering that performs reliably across a wide range of conditions. $\text{GSC}_\text{un}$ requires no tuning at $\alpha = 0$ and remains competitive with baselines in many cases. However, optimizing $(t, \alpha)$ yields substantial improvements, especially for $\text{GSC}_\text{n}$ and on complex graphs with skewed degree properties. Finally, the disagreement between unsupervised and supervised evaluation is not negligible, but rather highlights the difficulty of unsupervised model selection, and more specifically the mismatch between a considered internal evaluation index and the cluster structure present in a dataset. This finding leaves for future work the investigation of other more flexible internal indices and optimization strategies.

# 6 Conclusion

This work introduced the *Generalized Dirichlet Energy* (GDE) framework, which extends classical notions of graph smoothness to directed graphs through the use of arbitrary vertex measures and random walk dynamics. This leads to a principled construction of parametrized graph Laplacians that can capture both directional and structural properties without relying on teleportation or artificial symmetrization. Built on this foundation, we developed the *Generalized Spectral Clustering* (GSC) method, which leverages a parametrized vertex measure $\nu_{t,\alpha}$ to adapt to the geometry of directed graphs. The resulting model enables a smooth interpolation between local/global behavior and sparse/dense connectivity, offering a flexible and theoretically grounded approach to clustering in asymmetric and weakly connected settings.

Empirical evaluations on real-world point-cloud and network datasets show that GSC consistently performs well, often outperforming classical and teleportation-based baselines, under both internal and external evaluation metrics. Moreover, the proposed method has empirically demonstrated flexibility and robustness when dealing with various scenarios of structural biases that typically challenge spectral clustering methods. These results highlight the value of modeling diffusion via non-ergodic, structure-aware dynamics driven by carefully chosen vertex measures.

The versatility of the GDE framework opens up multiple avenues for future research, including its integration into semi-supervised learning, graph signal processing, and operator-based methods in non-reversible or directed settings. An especially promising direction is the automated or data-driven selection of $(t, \alpha)$, enabling fully adaptive graph inference across tasks and domains.

## Acknowledgments

H.S., G.D.-J., M.H., and A.K. were supported by the Industrial Analytics and Machine Learning (IdAML) Chair hosted at ENS Paris-Saclay, University Paris-Saclay. G.D.-J. was funded by a FMJH PhD scholarship. M.J. was funded by the International Centre for Mathematics and Computer Science (CImI) in Toulouse.

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

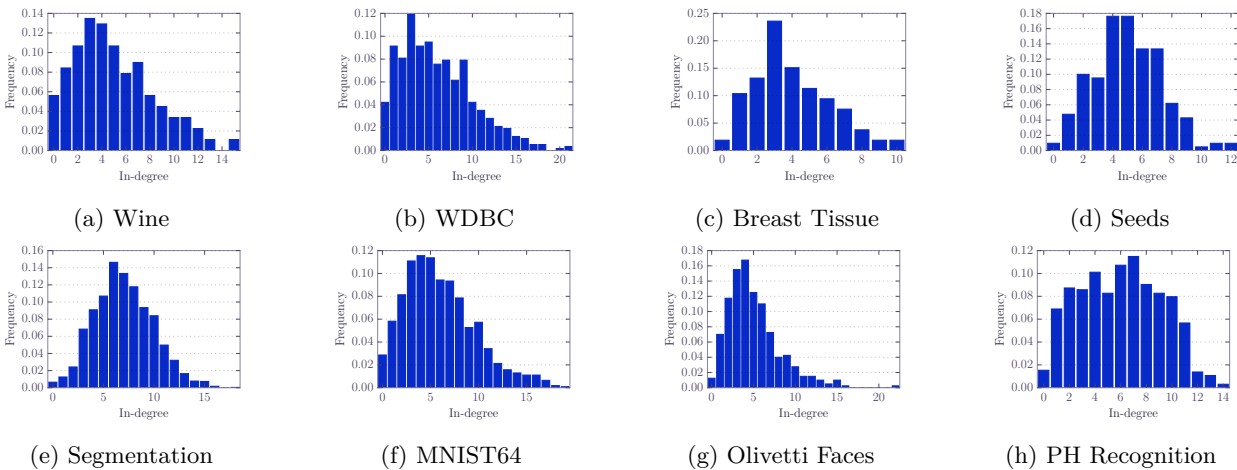

Figure 7: **In-degree distributions for the $K$-NN graphs.** Histograms of the vertex in-degrees for different datasets. Due to their construction as (directed) $K$-NN graphs, the distributions are not uniform.

Alistair Sinclair. Improved bounds for mixing rates of Markov chains and multicommodity flow. *Combinatorics, Probability and Computing*, 1(4):351–370, 1992.

Alexander Strehl and Joydeep Ghosh. Cluster ensembles—a knowledge reuse framework for combining multiple partitions. *Journal of Machine Learning Research*, 3(Dec):583–617, 2002.

Ulrike Von Luxburg. A tutorial on spectral clustering. *Statistics and Computing*, 17(4):395–416, 2007.

Dengyong Zhou, Jiayuan Huang, and Bernhard Schölkopf. Learning from labeled and unlabeled data on a directed graph. In *International Conference on Machine Learning*, pp. 1036–1043, 2005.

## A  Additional Information on the used datasets

Table 3 provides some statistics on the datasets used in the experiments: the number of samples $N$, the number of edges $|E|$, the graph connectivity properties (i.e. number of weakly-connected components and number of strongly-connected components), the number of data classes $K$, as well as statistics on the graph structure, such as the Gini coefficient of the in-degree distribution, the Reciprocity of edge weights, and the Cluster-Level reciprocity. The latter is a measure of how reciprocal the connections are between pairs of ground-truth clusters, and can provide insights into the suitability of the different clustering methods. For binary edge weights, all these measures lie in $[0, 1]$ and are defined as follows:

$$\text{Gini}(x) = \frac{\sum_{i=1}^{N} \sum_{j=1}^{N} |x_i - x_j|}{2N \sum_{i=1}^{N} x_i},$$

$$\text{Reciprocity}(\mathbf{W}) = \frac{\sum_{i,j} \mathbf{W}_{ij} \cdot \mathbf{W}_{ji}}{\sum_{i,j} \mathbf{W}_{ij}},$$

$$\text{CL Reciprocity}(\mathbf{W}, y) = \frac{1}{K(K-1)} \sum_{c \neq c'} \frac{2 \cdot \min\big(E(c, c'), E(c', c)\big)}{E(c, c') + E(c', c)},$$

where $E(c, c') = \sum_{i \in c, j \in c'} \mathbf{W}_{ij}$ is the number of directed edges from cluster $c$ to cluster $c'$, $y$ is a ground-truth cluster assignment, and $K$ is the total number of clusters. In addition, we provide in Fig. 7 the in-degree distributions of a subset of the $K$-NN graphs constructed from the different datasets. Due to their construction, those graphs have a fixed out-degree equal to $K$, but their in-degree distribution can be quite skewed. This motivates the use of flexible vertex measures that can adapt to such heterogeneity.

Table 3: **Dataset statistics.** For each real dataset, we report the number of samples $N$, the number of edges $|E|$, the number of classes $K$, the numbers of weakly and strongly connected components (#WCC and #SCC), the in-degree Gini coefficient, edge reciprocity, and cluster-level reciprocity. Network datasets are indicated by a '*'.

| Dataset | $N$ | $|E|$ | $K$ | #WCC | #SCC | Gini | Reciprocity | CL Reciprocity |
|---|---|---|---|---|---|---|---|---|
| Breast Tissue | 106 | 424 | 6 | 1 | 8 | 0.308 | 0.646 | 0.418 |
| * Email-Eu-Core | 1,005 | 25,571 | 42 | 20 | 203 | 0.539 | 0.718 | 0.542 |
| * Football | 115 | 1,226 | 12 | 1 | 1 | 0.040 | 1.000 | 0.894 |
| Iris | 150 | 750 | 3 | 2 | 13 | 0.321 | 0.635 | 0.283 |
| MNIST64 | 1,082 | 6,492 | 6 | 2 | 47 | 0.341 | 0.615 | 0.275 |
| Olivetti Faces | 400 | 2,000 | 40 | 4 | 38 | 0.331 | 0.661 | 0.039 |
| PH Recognition | 653 | 3,918 | 15 | 2 | 132 | 0.303 | 0.646 | 0.297 |
| * PolBlogs | 814 | 15,939 | 2 | 2 | 12 | 0.676 | 0.290 | 0.860 |
| * PolBooks | 105 | 882 | 3 | 1 | 1 | 0.328 | 1.000 | 1.000 |
| Seeds | 210 | 1,050 | 3 | 1 | 6 | 0.257 | 0.690 | 0.589 |
| Segmentation | 2,310 | 16,170 | 7 | 2 | 57 | 0.232 | 0.725 | 0.258 |
| WDBC | 569 | 3,414 | 2 | 1 | 52 | 0.382 | 0.478 | 0.796 |
| Wine | 178 | 890 | 3 | 1 | 18 | 0.371 | 0.575 | 0.196 |

***Directed Stochastic Block Models.*** In addition to the real-world datasets, we also evaluate our method on synthetic datasets generated from directed stochastic block models (DiSBM). These models allow us to control the level of reciprocity and degree imbalance in the graph, providing a more systematic evaluation of the clustering methods under different structural conditions. The DiSBM is defined by a block structure with specified probabilities of edge formation between and within blocks, which can be tuned to create various levels of connectivity and directionality. By varying these parameters, we can simulate scenarios ranging from highly reciprocal to highly asymmetric graphs, and assess how well GSC and other methods can recover the underlying cluster structure in each case. In particular, we consider a 'chain' DiSBM, where 3 clusters are connected in a chain-like structure, with edges predominantly directed from one cluster to the next. This setup allows us to evaluate the ability of the methods to capture directional relationships and recover the correct clustering in a non-reciprocal setting. It is defined as follows:

$$N = [500, 500, 500], \quad \boldsymbol{Q}_\rho = \begin{bmatrix} 0.1 & \rho & 0.01 \\ 0.01 & 0.1 & \rho \\ 0.01 & 0.01 & 0.1 \end{bmatrix},$$

where $N$ is the vector of cluster sizes and $\boldsymbol{Q}_\rho$ is the matrix of edge probabilities, with $\rho$ controlling the level of directionality. By default, $\rho$ is set to 0.15, creating a strong directional bias to the graph.

We also generate degree-corrected directed SBMs with block-specific power-law distributions. For each block $k$, we sample vertex propensities $\theta_i$ by drawing from a Pareto($\alpha_k$) distribution, normalizing to unit mean, then scaling by $\sigma_k$. The edge probability between vertices $i$ and $j$ is $p_{ij} = \min(p_{z_i z_j} \cdot \theta_i \theta_j, 1)$, where $z_i$ is vertex $i$'s block label and $p_{z_i z_j}$ is the baseline probability between blocks $z_i$ and $z_j$. For our experiments, we use three balanced blocks of 500 vertices with intra-block probability $p_{\text{intra}} = 0.05$ and inter-block probability $p_{\text{inter}} = 0.01$. Pareto exponents $(\alpha_1, \alpha_2, \alpha_3) = (1.8, 3.5, 3.5)$ yield a heavy-tailed first block, while scaling factors $(\sigma_1, \sigma_2, \sigma_3) = (2.5, 0.7, 0.7)$ give it substantially higher expected degree.

# B    Different approaches for vertex measures

In this section, we provide additional details on the different approaches for vertex measures that we have considered in our experiments. In particular, those vertex measures are:

- $\nu_{t,\alpha}$: the proposed vertex measure (see Section 4).

- $\nu_{\text{deg}}(\gamma)$: a degree-based vertex measure defined as $\nu_{\text{deg}}(\gamma)(i) = \gamma d_{\text{in}}(i) + (1 - \gamma)d_{\text{out}}(i)$. This measure interpolates between in-degree and out-degree, with $\gamma \in [0, 1]$ controlling the balance. Generally, $\gamma$ is set to 0.5.

- $\nu_{\text{unif}}$: the uniform vertex measure, where $\nu_{\text{unif}}(i) = 1/|\mathcal{V}|$ for all $i \in \mathcal{V}$.

- $\nu_{\text{Perron}}$[7]: the Perron vector of the random walk transition matrix $\mathbf{P}$, which corresponds to the stationary distribution of the natural random walk on the graph.

As stated in Section 3.3, the proposed vertex measure $\nu_{t,\alpha}$ recovers the uniform measure when $\alpha = 0$. Using $\nu_{\text{Perron}}$ (the stationary distribution) recovers the Directed Laplacians of Chung (2005). Unlike DSC+ that uses teleportation, the Perron vector may be supported on a subset of vertices in weakly connected graphs (Section 3.1), and thus may lead to degrading performance. The results in Table 4 and Table 5 show that $\nu_{t,\alpha}$ attains the strongest supervised performance while remaining competitive in the unsupervised setting. The measure $\nu_{\text{unif}}$ also performs well across both regimes. In contrast, $\nu_{\text{deg}}(\gamma)$ and $\nu_{\text{Perron}}$ show regime-dependent behavior: on the $K$-NN graphs, $\nu_{\text{deg}}(\gamma)$ is of essentially equivalent performance to $\nu_{t,\alpha}$ while the Perron measure is markedly weaker, particularly in its unnormalized form; on the network datasets the picture reverses, with $\nu_{\text{deg}}(\gamma)$ degrading substantially and the Perron measure remaining moderately competitive. Overall, these findings highlight the value of flexible, adaptive vertex measures and underscore the effectiveness of $\nu_{t,\alpha}$, while also validating alternative choices when tuning $(t, \alpha)$ is computationally expensive or when nonparametric measures are preferred. The caveat is that, except for $\nu_{\text{unif}}$, each parameter-free alternative is reliable in only one of the two regimes.

## C  Proofs

### C.1  Proof of Proposition 3.1

*Proof.* We begin by expanding the Dirichlet form:

$$
\begin{aligned}
\mathcal{D}_\nu^2(f) &= \sum_{i,j \in \mathcal{V}} \nu(i)p(i,j)[f(i) - f(j)]^2 \\
&= \sum_{i,j \in \mathcal{V}} \nu(i)p(i,j) \left( f(i)^2 + f(j)^2 - 2f(i)f(j) \right) \\
&= \sum_{i \in \mathcal{V}} \nu(i)f(i)^2 + \sum_{j \in \mathcal{V}} \left( \sum_{i \in \mathcal{V}} \nu(i)p(i,j) \right) f(j)^2 - 2 \sum_{i,j \in \mathcal{V}} \nu(i)p(i,j)f(i)f(j) \\
&= \sum_{i \in \mathcal{V}} \nu(i)f(i)^2 + \sum_{j \in \mathcal{V}} \xi(j)f(j)^2 - \left( \sum_{i,j \in \mathcal{V}} \nu(i)p(i,j)f(i)f(j) + \sum_{i,j \in \mathcal{V}} \nu(j)p(j,i)f(i)f(j) \right) \\
&= \langle f, \mathbf{D}_\nu f \rangle + \langle f, \mathbf{D}_\xi f \rangle - \langle f, (\mathbf{D}_\nu \mathbf{P} + \mathbf{P}^\mathsf{T} \mathbf{D}_\nu)f \rangle \\
&= \langle f, (\mathbf{D}_{\nu+\xi} - \mathbf{D}_\nu \mathbf{P} - \mathbf{P}^\mathsf{T} \mathbf{D}_\nu)f \rangle \\
&= \langle f, \mathbf{L}_\nu f \rangle \\
&= \langle f, \mathbf{D}_{\nu+\xi}^{-1} \mathbf{L}_\nu f \rangle_{\nu+\xi} \\
&= \langle f, \mathbf{L}_{\text{RW},\nu} f \rangle_{\nu+\xi}
\end{aligned}
$$

where we used the identity $\langle f, g \rangle_{\nu+\xi} = \langle f, (\mathbf{D}_\nu + \mathbf{D}_\xi)g \rangle$, and the definition of the generalized random walk Laplacian $\mathbf{L}_{\text{RW},\nu}$ (see Definition 3.2).

---

[7]In practice, having zero entries makes the definition of the normalized generalized Laplacians intractable, hence, we set those entries to a small positive value.

Table 4: **Comparison of vertex measures on UCI datasets.** (a-b) Unsupervised evaluation by optimizing CH: the first table shows CH scores, while the second shows the corresponding AMI scores. (c) Supervised evaluation by optimizing AMI directly. Optimized hyperparameters are shown in parentheses. Bold on shaded background marks the best-performing method per dataset, based on unrounded scores.

(a) UNSUPERVISED EVALUATION (CH SCORES | CH-OPTIMIZED)

| Dataset | $\nu_{t,\alpha}$ | | $\nu_{\deg}(\gamma)$ | | $\nu_{\text{unif}}$ | | $\nu_{\text{Perron}}$ | |
| --- | --- | --- | --- | --- | --- | --- | --- | --- |
| | $\text{GSC}_{\text{un}}(t,\alpha)$ | $\text{GSC}_{\text{n}}(t,\alpha)$ | $\text{deg-GSC}_{\text{un}}(\gamma)$ | $\text{deg-GSC}_{\text{n}}(\gamma)$ | $\text{uniform-GSC}_{\text{un}}$ | $\text{uniform-GSC}_{\text{n}}$ | $\text{perron-GSC}_{\text{un}}$ | $\text{perron-GSC}_{\text{n}}$ |
| Breast Tissue | **54.78** (1, 0.4) | 52.04 (10, 0.4) | 54.73 (0.9) | 50.57 (0.4) | 50.98 | 47.79 | 28.69 | 33.76 |
| Iris | 555.67 (0, 0.0) | **558.06** (7, 0.1) | 555.67 (0.0) | 555.67 (0.0) | 555.67 | 555.67 | 399.95 | 555.67 |
| MNIST64 | 139.69 (0, 0.0) | **161.54** (3, 0.5) | 141.14 (1.0) | 141.14 (1.0) | 139.69 | 139.69 | 6.02 | 29.80 |
| Olivetti Faces | 14.42 (0, 0.0) | **14.88** (17, 0.1) | 14.51 (0.4) | 14.61 (0.3) | 14.42 | 14.40 | 2.84 | 2.87 |
| PH Recognition | 436.85 (0, 0.0) | **438.12** (2, 0.1) | 436.85 (0.0) | 436.85 (0.0) | 436.85 | 436.85 | 17.78 | 9.22 |
| Seeds | 247.31 (0, 0.0) | 248.04 (3, 0.1) | 247.31 (0.0) | **248.60** (0.4) | 247.31 | 247.31 | 2.10 | 238.13 |
| Segmentation | 419.00 (0, 0.0) | **507.94** (12, 0.6) | 473.88 (0.6) | 473.88 (0.6) | 419.00 | 435.75 | 324.73 | 134.89 |
| WDBC | 257.69 (0, 0.0) | **258.74** (22, 0.6) | 257.69 (0.0) | 257.69 (0.0) | 257.69 | 257.69 | 1.84 | 247.74 |
| Wine | 70.37 (0, 0.0) | **70.44** (22, 0.1) | 70.37 (0.0) | **70.44** (1.0) | 70.37 | 70.37 | 1.72 | 69.97 |
| *PRB*(CH) | 0.96 | **0.99** | 0.97 | 0.97 | 0.95 | 0.95 | 0.24 | 0.58 |

(b) UNSUPERVISED EVALUATION (AMI SCORES | CH-OPTIMIZED)

| Dataset | $\nu_{t,\alpha}$ | | $\nu_{\deg}(\gamma)$ | | $\nu_{\text{unif}}$ | | $\nu_{\text{Perron}}$ | |
| --- | --- | --- | --- | --- | --- | --- | --- | --- |
| | $\text{GSC}_{\text{un}}(t,\alpha)$ | $\text{GSC}_{\text{n}}(t,\alpha)$ | $\text{deg-GSC}_{\text{un}}(\gamma)$ | $\text{deg-GSC}_{\text{n}}(\gamma)$ | $\text{uniform-GSC}_{\text{un}}$ | $\text{uniform-GSC}_{\text{n}}$ | $\text{perron-GSC}_{\text{un}}$ | $\text{perron-GSC}_{\text{n}}$ |
| Breast Tissue | 0.48 (1, 0.4) | 0.48 (10, 0.4) | 0.49 (0.9) | **0.49** (0.4) | 0.46 | 0.45 | 0.41 | 0.39 |
| Iris | **0.80** (0, 0.0) | 0.77 (7, 0.1) | **0.80** (0.0) | **0.80** (0.0) | **0.80** | **0.80** | 0.70 | **0.80** |
| MNIST64 | 0.91 (0, 0.0) | **0.95** (3, 0.5) | 0.86 (1.0) | 0.86 (1.0) | 0.91 | 0.91 | 0.05 | 0.37 |
| Olivetti Faces | 0.70 (0, 0.0) | 0.70 (17, 0.1) | 0.72 (0.4) | 0.71 (0.3) | 0.70 | **0.72** | 0.24 | 0.22 |
| PH Recognition | **0.56** (0, 0.0) | 0.56 (2, 0.1) | **0.56** (0.0) | **0.56** (0.0) | 0.56 | 0.56 | 0.27 | 0.09 |
| Seeds | 0.74 (0, 0.0) | 0.74 (3, 0.1) | 0.74 (0.0) | **0.75** (0.4) | 0.74 | 0.74 | 0.01 | 0.72 |
| Segmentation | 0.64 (0, 0.0) | 0.61 (12, 0.6) | 0.61 (0.6) | 0.61 (0.6) | 0.64 | **0.64** | 0.35 | 0.38 |
| WDBC | 0.68 (0, 0.0) | 0.62 (22, 0.6) | 0.68 (0.0) | 0.68 (0.0) | 0.68 | 0.68 | 0.00 | **0.69** |
| Wine | **0.86** (0, 0.0) | 0.85 (22, 0.1) | **0.86** (0.0) | 0.85 (1.0) | **0.86** | **0.86** | 0.00 | 0.82 |
| *PRB*(AMI) | **0.98** | 0.97 | 0.98 | 0.98 | 0.98 | 0.98 | 0.35 | 0.68 |

(c) SUPERVISED EVALUATION (AMI SCORES | AMI-OPTIMIZED)

| Dataset | $\nu_{t,\alpha}$ | | $\nu_{\deg}(\gamma)$ | | $\nu_{\text{unif}}$ | | $\nu_{\text{Perron}}$ | |
| --- | --- | --- | --- | --- | --- | --- | --- | --- |
| | $\text{GSC}_{\text{un}}(t,\alpha)$ | $\text{GSC}_{\text{n}}(t,\alpha)$ | $\text{deg-GSC}_{\text{un}}(\gamma)$ | $\text{deg-GSC}_{\text{n}}(\gamma)$ | $\text{uniform-GSC}_{\text{un}}$ | $\text{uniform-GSC}_{\text{n}}$ | $\text{perron-GSC}_{\text{un}}$ | $\text{perron-GSC}_{\text{n}}$ |
| Breast Tissue | 0.51 (21, 0.1) | **0.51** (8, 1.5) | 0.49 (0.9) | 0.50 (1.0) | 0.46 | 0.45 | 0.41 | 0.39 |
| Iris | 0.80 (0, 0.0) | **0.88** (25, 0.8) | 0.80 (0.0) | 0.80 (0.0) | 0.80 | 0.80 | 0.70 | 0.80 |
| MNIST64 | 0.91 (0, 0.0) | **0.97** (17, 0.9) | 0.91 (0.9) | 0.91 (0.9) | 0.91 | 0.91 | 0.05 | 0.37 |
| Olivetti Faces | 0.70 (0, 0.0) | **0.74** (2, 0.8) | 0.72 (0.5) | 0.74 (0.9) | 0.70 | 0.72 | 0.24 | 0.22 |
| PH Recognition | **0.56** (0, 0.0) | 0.56 (0, 0.0) | **0.56** (0.0) | **0.56** (0.0) | 0.56 | 0.56 | 0.27 | 0.09 |
| Seeds | 0.74 (0, 0.0) | 0.77 (2, 1.0) | **0.79** (1.0) | 0.77 (0.9) | 0.74 | 0.74 | 0.01 | 0.72 |
| Segmentation | 0.64 (0, 0.0) | **0.71** (24, 1.5) | 0.66 (0.9) | 0.66 (0.8) | 0.64 | 0.64 | 0.35 | 0.38 |
| WDBC | 0.68 (0, 0.0) | **0.73** (5, 0.8) | 0.71 (0.8) | 0.71 (0.6) | 0.68 | 0.68 | 0.00 | 0.69 |
| Wine | **0.86** (0, 0.0) | 0.86 (0, 0.0) | 0.86 (0.0) | 0.86 (0.0) | 0.86 | 0.86 | 0.00 | 0.82 |
| *PRB*(AMI) | 0.95 | **1.00** | 0.96 | 0.97 | 0.94 | 0.94 | 0.33 | 0.65 |

Table 5: **Comparison of vertex measures on network datasets.** (a-b) Unsupervised evaluation by optimizing GCH on Hellinger-embedded one-step random-walk profiles: the first table shows GCH scores, while the second shows the corresponding AMI scores. (c) Supervised evaluation by optimizing AMI directly. Optimized hyperparameters are shown in parentheses. Bold on shaded background marks the best-performing method per dataset, based on unrounded scores.

(a) UNSUPERVISED EVALUATION (GCH SCORES | GCH-OPTIMIZED)

| Dataset | $\nu_{t,\alpha}$ | | $\nu_{\deg}(\gamma)$ | | $\nu_{\text{unif}}$ | | $\nu_{\text{Perron}}$ | |
| --- | --- | --- | --- | --- | --- | --- | --- | --- |
| | $\text{GSC}_{\text{un}}(t,\alpha)$ | $\text{GSC}_{\text{n}}(t,\alpha)$ | $\text{deg-GSC}_{\text{un}}(\gamma)$ | $\text{deg-GSC}_{\text{n}}(\gamma)$ | $\text{uniform-GSC}_{\text{un}}$ | $\text{uniform-GSC}_{\text{n}}$ | $\text{perron-GSC}_{\text{un}}$ | $\text{perron-GSC}_{\text{n}}$ |
| DiSBM Chain | 26.45 (3, 0.2) | **30.56** (6, 1.2) | 26.46 (0.0) | 27.03 (1.0) | 26.44 | 27.03 | 26.44 | 26.92 |
| Deg-Corr | **42.85** (0, 0.0) | 42.76 (0, 0.0) | 4.22 (0.1) | 5.77 (0.0) | **42.85** | 42.76 | 4.20 | 42.76 |
| Email-Eu-Core | 3.37 (0, 0.0) | **3.58** (23, 0.0) | 2.29 (0.9) | 2.53 (0.2) | 3.37 | 3.47 | 3.20 | 3.04 |
| Football | **9.77** (0, 0.0) | **9.77** (0, 0.0) | **9.77** (0.0) | **9.77** (0.0) | **9.77** | **9.77** | **9.77** | **9.77** |
| PolBlogs | **44.98** (0, 0.0) | 44.19 (2, 0.1) | 9.61 (0.5) | 36.55 (0.9) | **44.98** | 37.81 | 44.75 | 6.90 |
| PolBooks | 9.73 (1, 0.3) | **9.91** (1, 1.7) | 3.21 (0.0) | 9.81 (0.0) | 9.69 | 9.69 | 9.69 | 9.81 |
| *PRB*(GCH) | 0.96 | **1.00** | 0.52 | 0.75 | 0.96 | 0.95 | 0.80 | 0.81 |

(b) UNSUPERVISED EVALUATION (AMI SCORES | GCH-OPTIMIZED)

| Dataset | $\nu_{t,\alpha}$ | | $\nu_{\deg}(\gamma)$ | | $\nu_{\text{unif}}$ | | $\nu_{\text{Perron}}$ | |
| --- | --- | --- | --- | --- | --- | --- | --- | --- |
| | $\text{GSC}_{\text{un}}(t,\alpha)$ | $\text{GSC}_{\text{n}}(t,\alpha)$ | $\text{deg-GSC}_{\text{un}}(\gamma)$ | $\text{deg-GSC}_{\text{n}}(\gamma)$ | $\text{uniform-GSC}_{\text{un}}$ | $\text{uniform-GSC}_{\text{n}}$ | $\text{perron-GSC}_{\text{un}}$ | $\text{perron-GSC}_{\text{n}}$ |
| DiSBM Chain | 0.73 (3, 0.2) | **1.00** (6, 1.2) | 0.73 (0.0) | 0.58 (1.0) | 0.73 | 0.58 | 0.73 | 0.57 |
| Deg-Corr | **0.99** (0, 0.0) | 0.98 (0, 0.0) | 0.07 (0.1) | 0.10 (0.0) | **0.99** | 0.98 | 0.07 | 0.98 |
| Email-Eu-Core | 0.37 (0, 0.0) | **0.39** (23, 0.0) | 0.27 (0.9) | 0.31 (0.2) | 0.37 | 0.39 | 0.35 | 0.26 |
| Football | **0.90** (0, 0.0) | **0.90** (0, 0.0) | **0.90** (0.0) | **0.90** (0.0) | **0.90** | **0.90** | **0.90** | **0.90** |
| PolBlogs | **0.54** (0, 0.0) | 0.52 (2, 0.1) | 0.05 (0.5) | 0.40 (0.9) | **0.54** | 0.38 | 0.53 | 0.12 |
| PolBooks | 0.55 (1, 0.3) | **0.57** (1, 1.7) | 0.14 (0.0) | 0.55 (0.0) | 0.57 | 0.57 | 0.57 | 0.55 |
| *PRB*(AMI) | 0.94 | **1.00** | 0.47 | 0.70 | 0.94 | 0.88 | 0.78 | 0.74 |

(c) SUPERVISED EVALUATION (AMI SCORES | AMI-OPTIMIZED)

| Dataset | $\nu_{t,\alpha}$ | | $\nu_{\deg}(\gamma)$ | | $\nu_{\text{unif}}$ | | $\nu_{\text{Perron}}$ | |
| --- | --- | --- | --- | --- | --- | --- | --- | --- |
| | $\text{GSC}_{\text{un}}(t,\alpha)$ | $\text{GSC}_{\text{n}}(t,\alpha)$ | $\text{deg-GSC}_{\text{un}}(\gamma)$ | $\text{deg-GSC}_{\text{n}}(\gamma)$ | $\text{uniform-GSC}_{\text{un}}$ | $\text{uniform-GSC}_{\text{n}}$ | $\text{perron-GSC}_{\text{un}}$ | $\text{perron-GSC}_{\text{n}}$ |
| DiSBM Chain | 0.73 (0, 0.0) | **1.00** (6, 1.2) | 0.73 (0.0) | 0.58 (0.9) | 0.73 | 0.58 | 0.73 | 0.57 |
| Deg-Corr | **0.99** (0, 0.0) | 0.98 (0, 0.0) | 0.07 (0.1) | 0.10 (0.7) | **0.99** | 0.98 | 0.07 | 0.98 |
| Email-Eu-Core | 0.38 (25, 0.1) | **0.41** (0, 1.7) | 0.27 (0.9) | 0.31 (0.2) | 0.37 | 0.39 | 0.35 | 0.26 |
| Football | **0.90** (0, 0.0) | **0.90** (0, 0.0) | **0.90** (0.0) | **0.90** (0.0) | **0.90** | **0.90** | **0.90** | **0.90** |
| PolBlogs | **0.54** (0, 0.0) | 0.52 (2, 0.1) | 0.05 (0.5) | 0.40 (0.9) | **0.54** | 0.38 | 0.53 | 0.12 |
| PolBooks | **0.58** (8, 0.3) | 0.57 (1, 1.7) | 0.14 (0.0) | 0.55 (0.0) | 0.57 | 0.57 | 0.57 | 0.55 |
| *PRB*(AMI) | 0.94 | **0.99** | 0.47 | 0.69 | 0.94 | 0.87 | 0.77 | 0.73 |

As for the second equation, we can see that

$$\mathcal{D}_\nu^2(f) = \sum_{i,j\in\mathcal{V}} \nu(i)p(i,j)[f(i)-f(j)]^2$$

$$= \frac{1}{2}\left(\sum_{i,j\in\mathcal{V}} \nu(i)p(i,j)[f(i)-f(j)]^2 + \sum_{i,j\in\mathcal{V}} \nu(i)p(i,j)[f(i)-f(j)]^2\right)$$

$$= \frac{1}{2}\left(\sum_{i,j\in\mathcal{V}} \nu(i)p(i,j)[f(i)-f(j)]^2 + \sum_{i,j\in\mathcal{V}} \nu(j)p(j,i)[f(j)-f(i)]^2\right)$$

$$= \frac{1}{2}\sum_{i,j\in\mathcal{V}} (\nu(i)p(i,j)+\nu(j)p(j,i))[f(i)-f(j)]^2.$$

Above, we used the fact that $[f(j)-f(i)]^2 = [f(i)-f(j)]^2$ to symmetrize the expression. This shows that the GDE can be expressed as a sum over pairs of vertices, weighted by the symmetrized vertex-edge measure $\nu(i)p(i,j)+\nu(j)p(j,i)$, which captures the mutual influence between vertices $i$ and $j$ in both directions. $\square$

## C.2 Proof of Proposition 3.2

*Proof.* Recall that $\mathbf{L}_\nu = \mathbf{D}_{\nu+\xi} - \mathbf{D}_\nu\mathbf{P} - \mathbf{P}^\mathsf{T}\mathbf{D}_\nu$. For self-adjointness in $\ell^2(\mathcal{V})$, we compute:

$$\langle f, \mathbf{L}_\nu g\rangle = \langle f, (\mathbf{D}_{\nu+\xi} - \mathbf{D}_\nu\mathbf{P} - \mathbf{P}^\mathsf{T}\mathbf{D}_\nu)g\rangle$$
$$= \langle f, \mathbf{D}_{\nu+\xi}g\rangle - \langle f, [\mathbf{D}_\nu\mathbf{P} + \mathbf{P}^\mathsf{T}\mathbf{D}_\nu]g\rangle.$$

The first term is symmetric in $f$ and $g$ since $\mathbf{D}_{\nu+\xi}$ is a diagonal operator. For the second term, we have:

$$\langle f, [\mathbf{D}_\nu\mathbf{P} + \mathbf{P}^\mathsf{T}\mathbf{D}_\nu]g\rangle = \sum_{i,j\in\mathcal{V}} f(i)g(j)(\nu(i)p(i,j)+\nu(j)p(j,i)) = \langle g, [\mathbf{D}_\nu\mathbf{P} + \mathbf{P}^\mathsf{T}\mathbf{D}_\nu]f\rangle,$$

where we used that $\nu(i)p(i,j)+\nu(j)p(j,i)$ is symmetric in $i$ and $j$. Combining both, gives $\langle f, \mathbf{L}_\nu g\rangle = \langle g, \mathbf{L}_\nu f\rangle$, which proves self-adjointness.

For positive semi-definiteness, we use the fact that $\mathbf{L}_\nu$ is self-adjoint and that the GDE is defined as $\mathcal{D}_\nu^2(f) = \langle f, \mathbf{L}_\nu f\rangle$. Since $\mathcal{D}_\nu^2(f) \geq 0$ for all $f$, it follows that $\langle f, \mathbf{L}_\nu f\rangle \geq 0$ for all $f$, which implies that $\mathbf{L}_\nu$ is positive semi-definite.

For the self-adjointness of $\mathbf{L}_{\mathrm{RW},\nu}$ in $\ell^2(\mathcal{V}, \nu + \xi)$, we can see that

$$\langle f, \mathbf{L}_{\mathrm{RW},\nu}g\rangle_{\nu+\xi} = \langle f, \mathbf{D}_{\nu+\xi}^{-1}\mathbf{L}_\nu g\rangle_{\nu+\xi}$$
$$= \langle f, \mathbf{L}_\nu g\rangle$$
$$= \langle g, \mathbf{L}_\nu f\rangle$$
$$= \langle g, \mathbf{D}_{\nu+\xi}^{-1}\mathbf{L}_\nu f\rangle_{\nu+\xi}$$
$$= \langle g, \mathbf{L}_{\mathrm{RW},\nu}f\rangle_{\nu+\xi}.$$

The same way, the quadratic form of $\mathbf{L}_{\mathrm{RW},\nu}$ is given by the Generalized Dirichlet energy, implying the positive semi-definiteness of $\mathbf{L}_{\mathrm{RW},\nu}$.

As for the normalized Laplacian $\overline{\mathbf{L}}_\nu = \mathbf{D}_{\nu+\xi}^{-1/2}\mathbf{L}_\nu\mathbf{D}_{\nu+\xi}^{-1/2}$, we can see that $\overline{\mathbf{L}}_\nu^\mathsf{T} = \overline{\mathbf{L}}_\nu$, which proves self-adjointness. For the positive semi-definiteness, we can see that $\langle f, \overline{\mathbf{L}}_\nu f\rangle = \langle \mathbf{D}_{\nu+\xi}^{-1/2}f, \mathbf{L}_\nu\mathbf{D}_{\nu+\xi}^{-1/2}f\rangle$, which is nonnegative since $\mathbf{L}_\nu$ is positive semi-definite. $\square$

## C.3 Proof of Proposition 3.3

*Proof.* As $\mathbf{P}$ is ergodic and $\mu$ is assumed to be a probability distribution, we have that $\nu_t = [\mathbf{P}^t]^\mathsf{T}\mu \to \pi$. Let $\varepsilon_t = \pi - \nu_t$ be the element-wise difference between $\nu_t$ and $\pi$. We have that $\|\varepsilon_t\| \xrightarrow[t\to\infty]{} 0$. Now, we can

express the GDE of a graph function $f$ as follows:

$$
\begin{aligned}
\mathcal{D}^2_{\nu_t}(f) &= \sum_{i,j \in \mathcal{V}} \nu_t(i) p(i,j) [f(i) - f(j)]^2 \\
&= \sum_{i,j \in \mathcal{V}} (\pi(i) - \varepsilon_t(i)) p(i,j) [f(i) - f(j)]^2 \\
&= \mathcal{D}^2_\pi(f) - \sum_{i,j \in \mathcal{V}} \varepsilon_t(i) p(i,j) [f(i) - f(j)]^2
\end{aligned}
\tag{32}
$$

Since $\|\varepsilon_t\|_1 \to 0$, it is clear that the second term vanishes. This result indicates that the GDE of a graph function $f$, associated with the transition matrix $\mathbf{P}$ and a parametrized measure $\nu_t$, is the sum of a quadratic form involving the usual unnormalized Laplacian $\mathbf{L}_\pi$ (the Dirichlet energy $\mathcal{D}^2_\pi(f)$) and a vanishing term. Note that this vanishing term can also be seen as a quadratic form. $\qquad\square$

### C.4 Proof of Proposition 4.1

*Proof.* We begin by expressing the flow from $S$ to $\bar{S}$:

$$
\begin{aligned}
q(S, \bar{S}) &= \sum_{i \in S, j \in \bar{S}} \nu(i) p(i,j) \\
&= \sum_{i,j \in \mathcal{V}} \nu(i) p(i,j) \chi_S(i) \chi_{\bar{S}}(j) \\
&= \sum_{i,j \in \mathcal{V}} \nu(i) p(i,j) \chi_S(i) (1 - \chi_S(j)) \\
&= \sum_{i,j \in \mathcal{V}} \nu(i) p(i,j) \chi_S(i) - \sum_{i,j \in \mathcal{V}} \nu(i) p(i,j) \chi_S(i) \chi_S(j) \\
\Rightarrow \quad q(S, \bar{S}) &= \sum_{i \in \mathcal{V}} \nu(i) \chi_S(i) - \sum_{i,j \in \mathcal{V}} \nu(i) p(i,j) \chi_S(i) \chi_S(j).
\end{aligned}
\tag{33}
$$

Similarly, the flow from $\bar{S}$ to $S$ is:

$$
\begin{aligned}
q(\bar{S}, S) &= \sum_{i \in \bar{S}, j \in S} \nu(i) p(i,j) \\
&= \sum_{i,j \in \mathcal{V}} \nu(i) p(i,j) \chi_{\bar{S}}(i) \chi_S(j) \\
&= \sum_{i,j \in \mathcal{V}} \nu(i) p(i,j) (1 - \chi_S(i)) \chi_S(j) \\
&= \sum_{i,j \in \mathcal{V}} \nu(i) p(i,j) \chi_S(j) - \sum_{i,j \in \mathcal{V}} \nu(i) p(i,j) \chi_S(i) \chi_S(j) \\
\Rightarrow \quad q(\bar{S}, S) &= \sum_{j \in \mathcal{V}} \left( \sum_{i \in \mathcal{V}} \nu(i) p(i,j) \right) \chi_S(j) - \sum_{i,j \in \mathcal{V}} \nu(i) p(i,j) \chi_S(i) \chi_S(j).
\end{aligned}
\tag{34}
$$

Now consider the Dirichlet form evaluated at the indicator vector $\chi_S$:

$$
\begin{aligned}
\mathcal{D}^2_\nu(\chi_S) &= \sum_{(i,j) \in \mathcal{E}} \nu(i) p(i,j) \left| \chi_S(i) - \chi_S(j) \right|^2 \\
&= \sum_{i \in \mathcal{V}} \nu(i) \chi_S(i) + \sum_{j \in \mathcal{V}} \left( \sum_{i \in \mathcal{V}} \nu(i) p(i,j) \right) \chi_S(j) - 2 \sum_{i,j \in \mathcal{V}} \nu(i) p(i,j) \chi_S(i) \chi_S(j) \\
\Rightarrow \quad \mathcal{D}^2_\nu(\chi_S) &= q(S, \bar{S}) + q(\bar{S}, S), \qquad \text{(by Eqs. 33 and 34).}
\end{aligned}
$$

$\qquad\square$

## C.5  Proof of Corollary 4.1

*Proof.* Let $\pi$ be the stationary distribution of the natural random walk on $\mathcal{G}$, i.e. $\pi(j) = \sum_{i \in \mathcal{V}} \pi(i)p(i,j)$ for all $j$. From the proof of Proposition 4.1, we have:

$$q(S, \bar{S}) - q(\bar{S}, S) = \sum_{i \in \mathcal{V}} \pi(i)\chi_S(i) - \sum_{j \in \mathcal{V}} \left( \sum_{i \in \mathcal{V}} \pi(i)p(i,j) \right) \chi_S(j)$$

$$= \sum_{i \in \mathcal{V}} \pi(i)\chi_S(i) - \sum_{j \in \mathcal{V}} \pi(j)\chi_S(j) = 0.$$

Thus, $q(S, \bar{S}) = q(\bar{S}, S)$, and by Proposition 4.1, we conclude that:

$$\mathcal{D}_\pi^2(\chi_S) = q(S, \bar{S}) + q(\bar{S}, S) = 2q(S, \bar{S}).$$

$\square$

## C.6  Proof of Theorem 4.1

*Proof.* This proof is divided into two parts. In the first part, we show that $\phi_\nu \geq \lambda_2/2$. In the second part, we show that $\phi_\nu \leq \sqrt{2\lambda_2}$.

We start by noting that the Rayleigh quotient of a function $f$ with respect to the normalized generalized Laplacian $\overline{\mathbf{L}}_\nu$ can be expressed as:

$$\mathcal{R}_{\overline{\mathbf{L}}_\nu}(f) = \frac{\langle f, \overline{\mathbf{L}}_\nu f \rangle}{\langle f, f \rangle} = \frac{\langle f, \mathbf{D}_{\nu+\xi}^{-1/2} \mathbf{L}_\nu \mathbf{D}_{\nu+\xi}^{-1/2} f \rangle}{\langle f, f \rangle} = \frac{\langle \mathbf{D}_{\nu+\xi}^{-1/2} f, \mathbf{L}_\nu \mathbf{D}_{\nu+\xi}^{-1/2} f \rangle}{\langle f, f \rangle}.$$

If we define $g = \mathbf{D}_{\nu+\xi}^{-1/2} f$, we can rewrite the Rayleigh quotient as:

$$\mathcal{R}_{\overline{\mathbf{L}}_\nu}(f) = \frac{\langle \mathbf{D}_{\nu+\xi}^{-1/2} f, \mathbf{L}_\nu \mathbf{D}_{\nu+\xi}^{-1/2} f \rangle}{\langle f, f \rangle} = \frac{\langle g, \mathbf{L}_\nu g \rangle}{\langle \mathbf{D}_{\nu+\xi}^{1/2} g, \mathbf{D}_{\nu+\xi}^{1/2} g \rangle} = \frac{\langle g, \mathbf{L}_\nu g \rangle}{\langle g, g \rangle_{\nu+\xi}},$$

where $\langle g, g \rangle_{\nu+\xi} = \langle g, \mathbf{D}_{\nu+\xi} g \rangle$. In particular, the second smallest eigenvalue $\lambda_2$ of $\overline{\mathbf{L}}_\nu$ can be characterized as:

$$\lambda_2 = \min_{f \perp \mathbf{D}_{\nu+\xi}^{1/2} \mathbf{1}} \mathcal{R}_{\overline{\mathbf{L}}_\nu}(f) = \min_{g \perp_{\nu+\xi} \mathbf{1}} \mathcal{R}_{\mathbf{L}_\nu}(g),$$

where the minimum is taken over all functions orthogonal to the constant function $\mathbf{1}$ for the inner product $\langle \cdot, \cdot \rangle_{\nu+\xi}$, i.e. $\sum_{i \in \mathcal{V}} (\nu(i) + \xi(i))g(i) = 0$. Throughout the proof, we will denote by $\text{vol}(S)$ the volume of a set $S$ with respect to the measure $\nu + \xi$, i.e. $\text{vol}(S) = \sum_{i \in S} (\nu(i) + \xi(i))$.

***Lower-bound.*** We start by showing that $\phi_\nu \geq \lambda_2/2$. Let $S^* = \arg\min_{S \subset \mathcal{V}} \phi_\nu(S)$ be the optimal partition that achieves the minimum conductance, and assume without loss of generality that $\text{vol}(S^*) \leq \text{vol}(\mathcal{V})/2$. Consider the indicator function $\chi_{S^*}$ associated with this partition. Define the function $f = \chi_{S^*} \text{vol}(\overline{S}^*) - \text{vol}(S^*)\chi_{\overline{S}^*}$, which is orthogonal to the constant function $\mathbf{1}$ (in the weighted inner product space), indeed:

$$\langle f, \mathbf{1} \rangle_{\nu+\xi} = \langle \chi_{S^*} \text{vol}(\overline{S}^*) - \text{vol}(S^*)\chi_{\overline{S}^*}, \mathbf{1} \rangle_{\nu+\xi} = \text{vol}(S^*)\text{vol}(\overline{S}^*) - \text{vol}(S^*)\text{vol}(\overline{S}^*) = 0.$$

In particular, by the Rayleigh quotient, $f$ is a valid function for the variational characterization of $\lambda_2$ of $\mathbf{L}_\nu$, and therefore can act as a second eigenvector. Notice that when $i, j$ are part of the same set (either $i, j \in S^*$ or $i, j \in \overline{S}^*$), we have $f(i) - f(j) = 0$, and when $i, j$ are part of different sets (i.e. $i \in S^*$ and $j \in \overline{S}^*$, or vice

versa), the difference is non-zero. Therefore, the generalized Dirichlet energy of $f$ can be computed as:

$$
\begin{aligned}
\mathcal{D}_\nu^2(f) &= \sum_{i,j \in \mathcal{V}} \nu(i)p(i,j)(f(i) - f(j))^2 \\
&= \sum_{i,j \in \mathcal{V}} \nu(i)p(i,j)(\chi_{S^*}(i) - \chi_{S^*}(j))^2 (\mathrm{vol}(S^*) + \mathrm{vol}(\overline{S}^*))^2 \\
&= \mathcal{D}_\nu^2(\chi_{S^*})(\mathrm{vol}(S^*) + \mathrm{vol}(\overline{S}^*))^2 \\
&= (q(S^*, \bar{S}^*) + q(\bar{S}^*, S^*))(\mathrm{vol}(S^*) + \mathrm{vol}(\overline{S}^*))^2 \qquad \text{(by Proposition 4.1)} \\
&= (q(S^*, \bar{S}^*) + q(\bar{S}^*, S^*)) \mathrm{vol}(\mathcal{V})^2.
\end{aligned}
$$

By definition of the conductance, it holds that $q(S^*, \bar{S}^*) + q(\bar{S}^*, S^*) = \phi_\nu(S^*) \mathrm{vol}(S^*)$. On the other hand, the norm of $f$:

$$
\begin{aligned}
\langle f, f \rangle_{\nu+\xi} &= \langle \chi_{S^*} \mathrm{vol}(\overline{S}^*) - \mathrm{vol}(S^*)\chi_{\overline{S}^*}, \chi_{S^*} \mathrm{vol}(\overline{S}^*) - \mathrm{vol}(S^*)\chi_{\overline{S}^*} \rangle_{\nu+\xi} \\
&= \sum_{i \in S^*} (\nu(i) + \xi(i)) \mathrm{vol}(\overline{S}^*)^2 + \sum_{i \in \overline{S}^*} (\nu(i) + \xi(i)) \mathrm{vol}(S^*)^2 \\
&= \mathrm{vol}(S^*) \mathrm{vol}(\overline{S}^*)(\mathrm{vol}(S^*) + \mathrm{vol}(\overline{S}^*)) \\
&= \mathrm{vol}(S^*) \mathrm{vol}(\overline{S}^*) \mathrm{vol}(\mathcal{V}).
\end{aligned}
$$

Plugging these two expressions into the Rayleigh quotient of $f$, we have:

$$
\mathcal{R}_{\mathbf{L}_\nu}(f) = \frac{\mathcal{D}_\nu^2(f)}{\langle f, f \rangle_{\nu+\xi}} = \frac{(q(S^*, \bar{S}^*) + q(\bar{S}^*, S^*)) \mathrm{vol}(\mathcal{V})^2}{\mathrm{vol}(S^*) \mathrm{vol}(\overline{S}^*) \mathrm{vol}(\mathcal{V})} = \frac{\phi_\nu(S^*) \mathrm{vol}(\mathcal{V})}{\mathrm{vol}(\overline{S}^*)}.
$$

Since $\mathrm{vol}(\overline{S}^*) \geq \mathrm{vol}(\mathcal{V})/2$, we have $\mathcal{R}_{\mathbf{L}_\nu}(f) \leq \phi_\nu(S^*) \mathrm{vol}(\mathcal{V})/(\mathrm{vol}(\mathcal{V})/2) = 2\phi_\nu(S^*)$. Since $\lambda_2$ is the minimum Rayleigh quotient over all functions orthogonal to the constant function, we have

$$
\lambda_2 \leq \mathcal{R}_{\mathbf{L}_\nu}(f) \leq 2\phi_\nu(S^*) = 2\phi_\nu.
$$

**Upper bound.** Let $f_2$ be the second generalized eigenvector of the pair $(\mathbf{L}_\nu, \mathbf{D}_{\nu+\xi})$, or equivalently, the second eigenvector of $\mathbf{L}_{\mathrm{RW},\nu}$, so that $\langle f_2, \mathbf{1} \rangle_{\nu+\xi} = 0$ and $\lambda_2 = \frac{\langle f_2, \mathbf{L}_\nu f_2 \rangle}{\langle f_2, f_2 \rangle_{\nu+\xi}}$. Assume that the vertices are ordered so that $f_2(1) \geq f_2(2) \geq \cdots \geq f_2(N)$; by definition, it holds that $\langle f_2, \mathbf{1} \rangle_{\nu+\xi} = 0$. We will assume that the graph has no self-loops. This implies that we can define level sets as:

$$
S_k = \{i \in \mathcal{V} : f_2(i) \geq f_2(k)\} = \{1, 2, ..., k\}.
$$

We define $r$ to be the largest index such that $\mathrm{vol}(S_r) \leq \mathrm{vol}(\mathcal{V})/2$, and define $g$ and $h$ by:

$$
g(i) = \begin{cases} f_2(i) - f_2(r) & \text{if } f_2(i) \geq f_2(r), \\ 0 & \text{otherwise.} \end{cases} \quad \text{and} \quad h(i) = \begin{cases} f_2(r) - f_2(i) & \text{if } f_2(i) < f_2(r), \\ 0 & \text{otherwise.} \end{cases}
$$

Note that $g$ and $h$ are nonnegative functions, and that $f_2 = g - h$. Moreover, $g$ is supported on $S_r$, while $h$ is supported on $\overline{S_r}$. Using the fact that $\lambda_2 = \mathcal{R}_{\mathbf{L}_\nu}(f_2)$, we have:

$$
\begin{aligned}
\lambda_2 &= \frac{\langle f_2, \mathbf{L}_\nu f_2 \rangle}{\langle f_2, f_2 \rangle_{\nu+\xi}} \\
&= \frac{\langle g, \mathbf{L}_\nu g \rangle + \langle h, \mathbf{L}_\nu h \rangle}{\langle g, g \rangle_{\nu+\xi} + \langle h, h \rangle_{\nu+\xi}} \qquad \text{(by the definition of } g \text{ and } h) \\
&\geq \min \left\{ \frac{\langle g, \mathbf{L}_\nu g \rangle}{\langle g, g \rangle_{\nu+\xi}}, \frac{\langle h, \mathbf{L}_\nu h \rangle}{\langle h, h \rangle_{\nu+\xi}} \right\} \qquad \text{(by the mediant inequality).}
\end{aligned}
$$

The mediant inequality is given for fractions $a/b$ and $c/d$ with $b, d > 0$ as:

$$
\frac{a+c}{b+d} \geq \min \left\{ \frac{a}{b}, \frac{c}{d} \right\} =: m,
$$

which comes from the fact that $a \geq mb$ and $c \geq md$, which implies that $a + c \geq m(b + d)$, and therefore $(a + c)/(b + d) \geq m$. Without loss of generality, we assume that $\mathcal{R}_{\mathbf{L}_\nu}(g) \leq \mathcal{R}_{\mathbf{L}_\nu}(h)$. In particular, we have that $\lambda_2 \geq \mathcal{R}_{\mathbf{L}_\nu}(g)$. Now, we can use the fact that $g$ is supported on $S_r$ to upper bound $\mathcal{R}_{\mathbf{L}_\nu}(g)$ as follows:

$$
\begin{aligned}
\mathcal{R}_{\mathbf{L}_\nu}(g) &= \frac{\langle g, \mathbf{L}_\nu g \rangle}{\langle g, g \rangle_{\nu + \xi}} \\
&= \frac{\sum_{i,j} \nu(i) p(i,j) (g(i) - g(j))^2}{\sum_i (\nu(i) + \xi(i)) g(i)^2} \\
&= \frac{\sum_{i,j} (\nu(i) p(i,j) + \nu(j) p(j,i)) (g(i) - g(j))^2}{2(\sum_i (\nu(i) + \xi(i)) g(i)^2)} \\
&= \frac{\sum_{i<j} (\nu(i) p(i,j) + \nu(j) p(j,i)) (g(i) - g(j))^2}{\sum_i (\nu(i) + \xi(i)) g(i)^2} \\
&= \frac{\sum_{i<j} (\nu(i) p(i,j) + \nu(j) p(j,i)) (g(i) - g(j))^2}{\sum_i (\nu(i) + \xi(i)) g(i)^2} \times \frac{\sum_{i<j} (\nu(i) p(i,j) + \nu(j) p(j,i)) (g(i) + g(j))^2}{\sum_{i<j} (\nu(i) p(i,j) + \nu(j) p(j,i)) (g(i) + g(j))^2} \\
&\geq \frac{\left( \sum_{i<j} (\nu(i) p(i,j) + \nu(j) p(j,i)) |g(i)^2 - g(j)^2| \right)^2}{\sum_i (\nu(i) + \xi(i)) g(i)^2 \times \sum_{i<j} (\nu(i) p(i,j) + \nu(j) p(j,i)) (g(i) + g(j))^2}.
\end{aligned}
$$

The last inequality is due to the Cauchy-Schwarz inequality, and the factor 2 comes from Proposition 3.1. For the denominator, we can use the fact that $(g(i) + g(j))^2 \leq 2(g(i)^2 + g(j)^2)$, so that:

$$
\begin{aligned}
\sum_{i<j} (\nu(i) p(i,j) + \nu(j) p(j,i)) (g(i) + g(j))^2 &\leq 2 \sum_{i<j} (\nu(i) p(i,j) + \nu(j) p(j,i)) (g(i)^2 + g(j)^2) \\
&= 2 \sum_{i,j} (\nu(i) p(i,j) + \nu(j) p(j,i)) g(i)^2 \\
&= 2 \sum_i (\nu(i) + \xi(i)) g(i)^2.
\end{aligned}
$$

As for the numerator, we can see that it can be expressed using the conductance of the level sets of $f_2$. Indeed, we have that:

$$
\begin{aligned}
\sum_{i<j} (\nu(i) p(i,j) + \nu(j) p(j,i)) |g(i)^2 - g(j)^2| &= \sum_{j=1}^N \sum_{i=1}^{j-1} (\nu(i) p(i,j) + \nu(j) p(j,i)) |g(i)^2 - g(j)^2| \\
&= \sum_{j=1}^N \sum_{i=1}^{j-1} (\nu(i) p(i,j) + \nu(j) p(j,i)) \sum_{k=i}^{j-1} (g(k)^2 - g(k+1)^2),
\end{aligned}
$$

where the last equality is due to the fact that we can write the telescopic sum $g(i)^2 = \sum_{k=i}^r (g(k)^2 - g(k+1)^2)$, and that $g(r+1) = 0$. Now, we can exchange the order of the summation to get:

$$
\begin{aligned}
\sum_{i<j} (\nu(i) p(i,j) + \nu(j) p(j,i)) |g(i)^2 - g(j)^2| &= \sum_{k=1}^{N-1} (g(k)^2 - g(k+1)^2) \sum_{j=k+1}^N \sum_{i=1}^k (\nu(i) p(i,j) + \nu(j) p(j,i)) \\
&= \sum_{k=1}^{N-1} (g(k)^2 - g(k+1)^2) (q(S_k, \bar{S}_k) + q(\bar{S}_k, S_k)).
\end{aligned}
$$

By definition of the generalized conductance, we have that for all $k$,

$$
q(S_k, \bar{S}_k) + q(\bar{S}_k, S_k) \geq \phi_\nu(S_k) \min\{\text{vol}(S_k), \text{vol}(\bar{S}_k)\}, \quad \text{and} \quad \phi_\nu(S_k) \geq \phi_\nu.
$$

Plugging this into the previous equation, we get:

$$
\lambda_2 \geq \frac{\left( \sum_{k=1}^{N-1} (g(k)^2 - g(k+1)^2) \phi_\nu(S_k) \min\{\text{vol}(S_k), \text{vol}(\bar{S}_k)\} \right)^2}{2 \langle g, g \rangle_{\nu + \xi}^2}.
$$

By the fact that $g$ is supported on $S_r$, we have that $\min\{\mathrm{vol}(S_k), \mathrm{vol}(\bar{S}_k)\} = \mathrm{vol}(S_k)$ for all $k \leq r$ and that $g(k) = 0$ for all $k > r$. Moreover, since $\phi_\nu(S_k) \geq \phi_\nu$ for all $k$, we have that:

$$\lambda_2 \geq \frac{\left(\sum_{k=1}^r (g(k)^2 - g(k+1)^2)\phi_\nu(S_k)\,\mathrm{vol}(S_k)\right)^2}{2\langle g, g\rangle_{\nu+\xi}^2} \geq \phi_\nu^2 \frac{\left(\sum_{k=1}^r (g(k)^2 - g(k+1)^2)\,\mathrm{vol}(S_k)\right)^2}{2\langle g, g\rangle_{\nu+\xi}^2}.$$

Using the fact that $g(r+1) = 0$ (by definition of $g$), and defining $\mathrm{vol}(S_0) = 0$, we can rewrite the numerator as:

$$\sum_{k=1}^r (g(k)^2 - g(k+1)^2)\,\mathrm{vol}(S_k) = g(1)^2\,\mathrm{vol}(S_1) + \sum_{k=2}^r g(k)^2(\mathrm{vol}(S_k) - \mathrm{vol}(S_{k-1})) - g(r+1)^2\,\mathrm{vol}(S_r)$$

$$= \sum_{k=1}^r g(k)^2(\mathrm{vol}(S_k) - \mathrm{vol}(S_{k-1}))$$

Moreover, by the definition of the volume, we have that $\mathrm{vol}(S_k) - \mathrm{vol}(S_{k-1}) = \nu(k) + \xi(k)$, so that:

$$\sum_{k=1}^r (g(k)^2 - g(k+1)^2)\,\mathrm{vol}(S_k) = \sum_{k=1}^r g(k)^2(\nu(k) + \xi(k)) = \langle g, g\rangle_{\nu+\xi}.$$

Putting this together, we have that:

$$\lambda_2 \geq \phi_\nu^2 \frac{\left(\sum_{k=1}^r (g(k)^2 - g(k+1)^2)\,\mathrm{vol}(S_k)\right)^2}{2\langle g, g\rangle_{\nu+\xi}^2} = \phi_\nu^2 \frac{\langle g, g\rangle_{\nu+\xi}^2}{2\langle g, g\rangle_{\nu+\xi}^2} = \frac{\phi_\nu^2}{2}.$$

This implies that $\phi_\nu \leq \sqrt{2\lambda_2}$, which concludes the proof of the upper bound. $\qquad\square$

