# OpenReview forum: "Generalized Dirichlet Energy and Graph Laplacians for Clustering Directed and Undirected Graphs"
_TMLR — Accepted by TMLR_

### Review · Reviewer_5qzM · 2025-11-28

**Summary Of Contributions:**

1. Given that traditional spectral clustering is a Dirichlet energy minimization problem on undirected graphs, the work firstly introduces the generalized Dirichlet energy framework that extends the original Dirichlet energy to measure the smoothness of arbitrary positive vertex measures.
2. A family of parametrized vertex measures is designed to capture local and global graph dynamics via random walks.
3. Generalized spectral clustering is proposed by substituting parametrized vertex measure into the generalized Dirichlet energy minimization framework, which is applicable to both undirected and directed graphs.
4. The approach is validated on directed knn graphs constructed from UCI datasets.

Overall, the motivation is reasonable and the proposed method is technically sound.

**Additional Comments:**

On datasets like Wdbc, Seeds, and Segmentation, hyperparameter combinations with large CH values yield poor AMI, indicating the model fails to assign proper clustering assignments.

**Audience:**

Yes

**Audience Explanation:**

spectral clustering on directed graphs is a fundamental research topic in machine learning.

**Claims And Evidence:**

No

**Claims Explanation:**

The claims are theoretically well-supported, but don't align well with the experiments.
1. The motivation arises from "In many modern applications, such as web graphs, citation networks, and information diffusion in social or transportation systems, ...", but the experiments are entirely based on graphs constructed from UCI (general tabular) datasets via a kernel function.
2. Connectivity is a well-defined concept in graph theory. While this work claims to support weakly connected digraphs, no empirical results are provided to analyze how connectivity affects clustering results.

**Requested Changes:**

1. Provide empirical results on "modern applications" to  comply with the motivation
2. If feasible, analyse how the graph connectivity affects the clustering algorithm

---

### Review · Reviewer_RMro · 2025-12-09

**Summary Of Contributions:**

This paper introduces the Generalized Dirichlet Energy (GDE) framework, which extends the classical Dirichlet energy to handle both directed and undirected graphs by incorporating arbitrary positive vertex measures and Markov transition matrices. The authors propose a novel energy functional that does not require reversibility or ergodicity, thus avoiding the pitfalls of traditional methods that rely on symmetrization or teleportation. A key contribution is the development of the Generalized Spectral Clustering (GSC) method, which leverages a parametrized vertex measure to capture graph directionality and density. The method avoids commonly used heuristics like teleportation (PageRank-style smoothing) or symmetrization, which often distort directional information in digraphs.

Strengths:
- The GDE framework is mathematically sound and unifies several prior constructions under a broader variational principle.
- The connection between the energy, Laplacian operators, and random-walk-based cuts is clearly derived .
- Empirical validation on real-world point-cloud datasets demonstrates consistent gains over multiple baselines, especially on asymmetric or weakly connected digraphs.

Weaknesses:
- The theoretical implications of using non-stationary measures for spectral clustering remain underdeveloped. In particular, spectral convergence or stability guarantees under this generalization are absent.
- The paper does not adequately address when or why the classical Dirichlet energy fails on directed graphs beyond citing ergodicity concerns—yet many real-world digraphs are strongly connected (hence ergodic), raising the question of necessity.
- The paper lacks a detailed discussion on the computational complexity of the GSC method, particularly in comparison to other spectral clustering techniques.
- The sensitivity analysis of the hyperparameters t and α is limited to specific datasets, and the generalizability of the findings is not fully explored.

**Audience:**

Yes

**Audience Explanation:**

Yes, TMLR's audience includes researchers in graph machine learning, spectral methods, and geometric data analysis. Directed graph clustering remains a persistent challenge, and the community has long grappled with the tension between preserving directionality and enabling spectral techniques. This work offers a principled, theoretically grounded alternative to prevailing heuristics (teleportation, symmetrization), which many will find valuable.
Moreover, the generalized Dirichlet energy connects to broader themes in graph signal processing, diffusion geometry, and operator learning—areas of high interest to TMLR readers.

**Broader Impact Concerns:**

The paper is theoretical/methodological and poses no immediate ethical risks.

**Claims And Evidence:**

Yes

**Claims Explanation:**

The core mathematical claims are correct and properly derived. The link between GDE and graph cuts is established via the indicator-function analysis, and the recovery of classical Laplacians when ν=π is verified.
The empirical claims—that GSC “consistently outperforms existing spectral clustering approaches”—are plausible but overstated. The experiments are well-structured: unsupervised model selection via Calinski–Harabasz (CH), post-hoc AMI evaluation, and comparison to strong baselines (DSC+, DI-SIM, symmetrized SC). Results show GSCn often achieves the best CH scores (avg. rank 1.5), and GSCun leads in AMI (avg. rank 2.0).
However, critical issues undermine full confidence:
- The datasets are all small to moderate in size (max N=2310), limiting evidence of scalability.
- The KNN construction uses unweighted, asymmetric edges, which is atypical—most modern pipelines use Gaussian or reciprocal KNN (e.g., mutual KNN) to reduce noise. The sensitivity to this choice is unexplored.
- The CH index is known to favor compact, spherical clusters; its suitability is asserted but not validated (e.g., via cluster shape diagnostics). In datasets like Olivetti Faces or MNIST64, where clusters are highly non-convex, CH may be misleading—yet GSC still relies on it for model selection.
- There is no ablation on the role of α vs. t . Is the gain due to temporal diffusion, reweighting, or both?
Thus, while the evidence is convincing for proof-of-concept, it falls short of demonstrating robust superiority across diverse directed graph regimes.

**Requested Changes:**

- Provide concrete examples (theoretical or empirical) where ergodic digraphs still fail under classical Dirichlet energy (e.g., due to poor mixing, skewed stationary distribution). Without this, the motivation appears speculative.
- Strengthen empirical evaluation:
	- Include at least one large-scale digraph to assess scalability.
	- Test on synthetic directed stochastic block models (diSBM) with controllable directionality strength; show GSC's advantage grows with asymmetry.
	- Replace unweighted KNN with Gaussian-weighted or mutual KNN to align with standard practice.
- Analyze failure modes: In Figure 1, some datasets (e.g., PH Recognition) show poor AMI even at best CH. Discuss why—is it due to CH mismatch, intrinsic cluster ambiguity, or method limitation?
- Provide a detailed analysis of the computational complexity of the GSC method, including comparisons with existing spectral clustering techniques.
- Ablate α and t : Show separate heatmaps for fixed α=1 (varying t ) and fixed t=0 (varying α ) to disentangle their effects.
- Since the intro mentions Cucuringu et al. (2020), briefly contrast GSC's density-based goal with flow-imbalanced cuts—when is each preferable?

---

### Review · Reviewer_wW1d · 2026-03-06

**Summary Of Contributions:**

The paper motivates the need for clustering methods for directed graph, which avoid common workarounds like symmetrization (losing directionality) and teleporting random walks (altering topology and potentially masking structure).

The main technical proposal is a generalized Dirichlet energy over an arbitrary positive edge measure $q(i,j)$, specializing to $q(i,j)=\nu(i)p(i,j)$ for a positive vertex measure $\nu$ and a Markov transition matrix $P$. The authors derive a family of generalized Laplacians built from $\nu$, its 'incoming measure' $\xi=P^\top \nu$, and symmetrized terms $D_\nu P + P^\top D_\nu$, and show the generalized energy equals a quadratic form in the corresponding Laplacian, thus yielding a self-adjoint operator in an appropriate inner product. Then, they introduce a parameterized vertex measure $\nu_{t,\alpha}$ obtained by diffusing a uniform distribution for $t$ steps (via $P^t$) and applying an elementwise exponent $\alpha$, interpreting $t$ as local and global diffusion scale and $\alpha$ as a 'sharpness' reweighting parameter. Finally, they instantiate generalized spectral clustering (GSC): compute the generalized Laplacian for a chosen $(t,\alpha)$, take the $k$ smallest-eigenvalue eigenvectors, cluster rows with $k$-means++, and return the partition. Empirically, they evaluate on UCI point-cloud datasets by constructing unweighted directed KNN graphs with $K=\lceil \log N\rceil$. They tune each method by Calinski–Harabasz (CH) as an internal index and report AMI post hoc. They compare against DSC+ (teleporting random walk), DI-SIM variants, and symmetrized spectral clustering (SC).

---
Strength:
1. The GDE to Laplacian pipeline is clean: defining an energy as $\sum q(i,j)(f(i)-f(j))^2$ and showing it equals a symmetric quadratic form provides a principled route to spectral methods, and Proposition 3.1 provides the key identity connecting energy and operator.
2. The parameterization $\nu_{t,\alpha}$ is intuitive and plausibly useful in practice; the paper articulates the 'diffusion time' and 'sharpness' interpretations explicitly, and links $\nu_t$ to $\pi$ in the ergodic case.

Weakness:
The computational implications of sweeping $(t,\alpha)$ are not discussed in depth, e.g., how $P^t$ is computed efficiently for $t\le 25$ across datasets; whether they reuse computations across $t$; sensitivity to graph sparsity. Yet the paper argues teleportation densifies graphs and increases cost, so a symmetric runtime comparison result is needed.

**Audience:**

Yes

**Audience Explanation:**

Yes. Researchers who work on graph ML, spectral methods, and clustering on directed graphs would likely care about this paper because it proposes a unified Dirichlet-energy/Laplacian framework that covers both directed and undirected settings (Defs. 3.1–3.2) and provides a clean random-walk cut interpretation for the objective (Prop. 4.1). That said, broader interest across TMLR will depend on stronger empirical evidence beyond the current evaluation setup (directed KNN graphs from UCI datasets).

**Claims And Evidence:**

Yes

**Claims Explanation:**

Math derivations are mostly accurate and presented, with proofs for key propositions.

The simulation results consistently indicate the superior performance of GSC.

**Requested Changes:**

1. Justify necessity of the “arbitrary vertex measure” generalization: clearly state what new capability is enabled (beyond $\nu = \pi$), when it helps, and provide guidance for choosing $\nu$ (or show failure modes).
2. Add numerical test on the choice of $\nu$ (uniform / degree / stationary / $\nu_{t, \alpha}$), sensitivity to $K$.
3. Add runtime and compare computational cost to other baselines in the experiments.

---

### Decision · Action_Editor_Yuxm · 2026-06-08

**Recommendation:** Accept as is

**Audience:**

Yes

**Audience Explanation:**

The topic is relevant to the ML community.

**Claims And Evidence:**

Yes

**Claims Explanation:**

The paper is accepted. It presents a significant and well-executed theoretical advance for a long-standing problem in graph machine learning. The reviewers were in clear agreement that the Generalized Dirichlet Energy framework is a valuable contribution. The authors' diligent and effective response to the initial reviews greatly strengthened the paper by adding crucial justifications, experiments, and theoretical backing. While the empirical validation has some limitations, these are clearly articulated and do not undermine the importance of the core theoretical work. The paper meets the TMLR criteria for claims, evidence, and audience, and represents a solid contribution to the field.